The wings before the bird: an evaluation of flapping-based locomotory hypotheses in bird antecedents

Dececchi T. Alexander 1 td50@queensu.ca
Larsson Hans C.E. 2
Habib Michael B. 3 4
1 Department of Geological Sciences, Queens University , Kingston, Ontario , Canada
2 Redpath Museum, McGill University , Montreal, Quebec , Canada
3 Keck School of Medicine of USC, Department of Cell and Neurobiology, University of Southern California , Los Angeles, California , United States
4 Dinosaur Institute, Natural History Museum of Los Angeles , Los Angeles, CA , United States
Farke Andrew
Electronic publication date: 2016 Jul 7
Publication date: 2016
Volume: 4
Electronic Location ID: e2159
Received 2016 Jan 23; Accepted 2016 May 27
Copyright: © 2016 Dececchi et al.
Copyright year: 2016
Copyright holder: Dececchi et al.
License: This is an open access article distributed under the terms of the Creative Commons Attribution License, which permits unrestricted use, distribution, reproduction and adaptation in any medium and for any purpose provided that it is properly attributed. For attribution, the original author(s), title, publication source (PeerJ) and either DOI or URL of the article must be cited.
License URL: https://creativecommons.org/licenses/by/4.0/

Keywords: Flight, WAIR, Maniraptora, Macroevolution, Theropoda, Flap running, Flight stroke

Funding: The authors received no funding for this work.

==============================
Background: Powered flight is implicated as a major driver for the success of birds. Here we examine the effectiveness of three hypothesized pathways for the evolution of the flight stroke, the forelimb motion that powers aerial locomotion, in a terrestrial setting across a range of stem and basal avians: flap running, Wing Assisted Incline Running (WAIR), and wing-assisted leaping.

Methods: Using biomechanical mathematical models based on known aerodynamic principals and in vivo experiments and ground truthed using extant avians we seek to test if an incipient flight stroke may have contributed sufficient force to permit flap running, WAIR, or leaping takeoff along the phylogenetic lineage from Coelurosauria to birds.

Results: None of these behaviours were found to meet the biomechanical threshold requirements before Paraves. Neither was there a continuous trend of refinement for any of these biomechanical performances across phylogeny nor a signal of universal applicability near the origin of birds. None of these flap-based locomotory models appear to have been a major influence on pre-flight character acquisition such as pennaceous feathers, suggesting non-locomotory behaviours, and less stringent locomotory behaviours such as balancing and braking, played a role in the evolution of the maniraptoran wing and nascent flight stroke. We find no support for widespread prevalence of WAIR in non-avian theropods, but can’t reject its presence in large winged, small-bodied taxa like Microraptor and Archaeopteryx.

Discussion: Using our first principles approach we find that “near flight” locomotor behaviors are most sensitive to wing area, and that non-locomotory related selection regimes likely expanded wing area well before WAIR and other such behaviors were possible in derived avians. These results suggest that investigations of the drivers for wing expansion and feather elongation in theropods need not be intrinsically linked to locomotory adaptations, and this separation is critical for our understanding of the origin of powered flight and avian evolution.

Introduction

Evolution of powered flight in vertebrates was a key innovation that spurred the evolutionary success of birds, bats, and pterosaurs (Sears et al., 2006; Butler et al., 2009; Benson & Choiniere, 2013). Of the three radiations, the theropod to bird transition has garnered the most interest and scholarship due to the higher quality of the fossil record documenting the origin and refinement of their flight including: the evolution of feathers, reduced body size, an avian-like physiology and respiration, elongate forelimbs, and modifications of the pectoral and forelimb musculoskeletal system (Baier, Gatesy & Jenkins, 2007; Codd et al., 2008; Dececchi & Larsson, 2009; Dececchi & Larsson, 2013; Makovicky & Zanno, 2011; Benson & Choiniere, 2013; Brusatte et al., 2014; Xu et al., 2014). Despite the wealth of fossil evidence documenting this transition deducing the origin and subsequent evolution of the flight stroke, a biomechanical innovation that permitted aerial locomotion, remains elusive.

The flight stroke of extant birds traces a complex ellipsoidal path that is controlled by derived muscle origins and insertions and modified shoulder, elbow, and wrist joints and ligaments (Gatesy & Baier, 2005). Many antecedent functions of the flight stroke have been proposed. These include a raptorial function of the forelimbs for fast prey capture (Ostrom, 1974), behavioural precursors such as courtship, balance, or warning displays (Fowler et al., 2011; Foth, Tischlinger & Rauhut, 2014), as well as locomotory functions (Caple, Balda & Willis, 1983; Dial, 2003; Chatterjee & Templin, 2007).

Powered flight differs from gliding flight in that it uses active flapping to generate thrust. Some models of the origin of avian flight propose antecedents living in trees and deriving the flight stroke from a parachuting or gliding stage (Chatterjee & Templin, 2004; Alexander et al., 2010; Dyke et al., 2013) based primarily on the observation that many modern arboreal tetrapods perform similar behaviors (Dudley et al., 2007; Evangelista et al., 2014). Yet nearly all stem avians have hindlimb morphologies that compare most closely to extant cursorial tetrapods (Dececchi & Larsson, 2011) and a multivariate analysis of limb element lengths recovered the earliest birds as most similar to extant terrestrial foragers (Bell & Chiappe, 2011; Mitchell & Makovicky, 2014). The only theropod taxa that may diverge from this are Scansoriopterygidae, a clade known from four small, fragmentary specimens, but presenting intriguing and radically divergent morphologies from other maniraptoran theropods. Notably, when preserved, they possess large pedal and manual phalangeal indices, a reduced crural index, a reduced hindlimb length, and reduced limb integument not seen in avian antecedents, including paravians (Glen & Bennett, 2007; Bell & Chiappe, 2011; Dececchi & Larsson, 2011; Dececchi, Larsson & Hone, 2012). One scansoriopterygid may even possess a skin patagium that may have functioned as an airfoil (Xu et al., 2015). These putative gliding structures are extremely divergent from other theropods and likely represent a convergent pathway to becoming volant.

Of all the models for the origin of the flight stroke from a terrestrial life history two major categories exist: those that have locomotory functional aspect are flap running (Burgers & Chiappe, 1999), wing assisted incline running or WAIR (Dial, 2003), and vertical leaping (Caple, Balda & Willis, 1983). Behaviors in the second category are non-locomotory behaviors, such as balancing during prey capture (Fowler et al., 2011) and braking during high-speed turns (Schaller, 2008). The three stringent locomotory behaviours (WAIR, flap running and vertical leaping) are variations on a proto-flight stroke assisting in force generation to increase ground and launch velocities (Burgers & Chiappe, 1999) or to assist in ascending steep inclines to facilitate escape to elevated refuges such as into trees or up inclined rock faces (Dial, 2003). All three are present throughout much of extant bird diversity and have been areas of research into the possible pathways for the origins of powered flight.

WAIR is a behaviour observed primarily as a means of predator escape, especially in pre-flight capable juveniles (Tobalske & Dial, 2007; Dial, Jackson & Segre, 2008; Jackson, Segre & Dial, 2009; Heers & Dial, 2012; Heers, Dial & Tobalske, 2014). This has been suggested to provide a series of functional and morphological stages using immature age classes of extant individuals as proxies for transitional evolutionary stages from basal coelurosaurs to volant birds (Dial, Randall & Dial, 2006; Heers & Dial, 2012). This has been most thoroughly studied in the Chukar partridge (Alectornis chukar, hereafter referred to as Chukars), though work has been done in other extant birds such as the Brush Turkey (Alectura lathami) and Peafowl (Pavo cristatus) (Heers & Dial, 2015). At the earliest juvenile stages Chukars (0–5 days post hatching [dph] and < 20 g) either crawl or asymmetrically flap their wings to produce forces of approximately 6–10% of their body weight (Jackson, Segre & Dial, 2009; Heers, Tobalske & Dial, 2011; Heers, Dial & Tobalske, 2014) to ascend inclines of less than 65°, slightly greater than the level that they can ascend using their legs alone (55–60°) (Bundle & Dial, 2003; Dial, Randall & Dial, 2006). At these low angles, the primary locomotory forces are generated from the hindlimbs but this changes when higher angles are attempted (Bundle & Dial, 2003). To ascend to sub vertical angles, juvenile and older individuals must produce forces equaling a minimum of 50% of their body weight (Dial & Jackson, 2011). Larger birds with masses greater than 0.8 kg such as adult Brush Turkeys or Peafowl struggle to WAIR at this level (Dial & Jackson, 2011; Heers & Dial, 2015). Low angle WAIR has been hypothesized to be present throughout Coelurosauria and sub vertical WAIR minimally at Paraves (Dial, 2003; Heers & Dial, 2012; Heers, Dial & Tobalske, 2014).

Vertical leaping (both from the ground and perches) begins as an effectively ballistic process in flying animals, initiated by the hindlimbs in birds (Heppner & Anderson, 1985; Bonser & Rayner, 1996; Earls, 2000; Tobalske, Altshuler & Powers, 2004), bats (Schutt et al., 1997; Gardiner & Nudds, 2011), and insects (Nachtigall & Wilson, 1967; Nachtigall, 1968; Nachtigall, 1978; Schouest, Anderson & Miller, 1986; Trimarchi & Schneiderman, 1995; Dudley, 2002). Immediately after the ballistic phase is initiated, the wings are engaged for the climb out phase of launch. Leaping takeoffs are common among small to medium sized birds (Provini et al., 2012) but are also present in many larger birds including Turkeys (Tobalske & Dial, 2000), Peafowl (Askew, 2014), Tinamou (Silveira et al., 2001) as well as herons, storks, eagles, and vultures) (TA Dececchi and MB Habib, 2015, personal observations). The largest living flying birds, Kori bustards, are documented to use a very short run before launch (Prozesky, 1970), though large captive specimens have demonstrated a true leaping takeoff, as well (MB Habib, 2014, personal observations). Caple, Balda & Willis (1983) proposed as a model for the origin of flight in birds, especially in smaller taxa. Flap-running is used in some extant birds, especially semi aquatic species, to accelerate to takeoff speeds whether starting from a water or land launch (though mostly associated with compliant surfaces, e.g. water–see Earls, 2000).

Here we present biomechanical models to test when and if a flight stroke may have contributed to flap running, WAIR, or leaping takeoff along the phylogenetic lineage from Coelurosauria to birds and if these models coincide with the evolution of pennaceous feathers and musculoskeletal adaptations for flight. Our goal is to take evolutionary narratives about pathways to flight origins and evaluate them using quantitative, mechanical models derived from living birds. Although feathery integument is likely to have been a synapomorphy for all dinosaurs and perhaps even all ornithodirans (Godefroit et al., 2014 but see Barrett, Evans & Campione, 2015), the evolution of pennaceous forelimb and hindlimb feathers has been hypothesized to have been driven by selection for locomotion (Burgers & Chiappe, 1999; Xu et al., 2003; Dial, Randall & Dial, 2006; Heers, Tobalske & Dial, 2011). Thus we set up a testing regime to determine if non-avian theropods could produce biomechanical values that fit within the realms of those measured in modern animals exhibiting these behaviors, and if is there a decoupling of the timing of the success in these behaviours from the origin of previous proposed flight related traits.

Materials and Methods

Due to uncertainty regarding soft tissues in fossil organisms, some variables were treated as constants in the taxa modeled and based on values for extant birds. These include feather material properties, arrangement and muscle power. Using these values provided conservative estimates in the sense that they would yield more capable performances for taxa that may lie near biomechanical thresholds. Wing feather arrangements for some fossils appear to be similar to modern birds (Elżanowski, 2002; Xu et al., 2003; Foth, Tischlinger & Rauhut, 2014) though for some taxa this has been disputed (Xu, Zheng & You, 2010; Longrich et al., 2012).

A greater source of uncertainty and debate is fraction of forelimb muscle mass that is due to the M. pectoralis and its potential power output. Extant birds have extremely large wing muscles, as a proportion to their bodyweight (Marden, 1987). The mass of M. pectoralis for birds’ ranges between 10–20% of total body mass (Greenewalt, 1975; Askew, Marsh & Ellington, 2001), and total flight muscle fractions for birds can reach 40% (Hartman, 1961; Greenewalt, 1962). This is significantly larger than that estimated in non-avian theropods or early birds. For example, Archaeopteryx’s pectoral muscles are estimated at only 0.5% of its body mass (Bock, 2013) with the entire forelimb (including bone and all other tissues) at 11–14% (Allen et al., 2013). For our analysis, we calculated values for power available from the forelimb and hindlimb based on the assumption that non-avian theropods had forelimb muscle mass fractions of 10% their total mass and that hindlimb muscle mass fractions were 30% of total mass. These values are likely significant overestimations for non-paravians pectoral regions, but the pelvic region values are within the range previous estimated for non-avian maniraptorans (Allen et al., 2013), whose estimates do not include the M. caudofemoralis. The pectoral muscle values we assigned are similar to estimates of pectoral region mass in Microraptor and Archaeopteryx, though those estimates are based on the entire pectoral region tissues (except feathers) and thus the relative mass of the pectoral musculature is likely smaller.

Yet power and muscle mass may not be the main determinant for the use of wings as locomotory structures. Jackson, Tobalske & Dial (2011) estimated that pigeons, with approximately 20% of their body mass as pectoralis muscles, only used approximately 10% of their mass-specific power for low angle WAIR. Further, it has been suggested that power output itself may not determine flight ability, but lift to power ratio (Marden, 1987). For this analysis we have assumed extant bird power productions and metabolic capacities for short “burst” activities for non-avian theropods and early birds. Although paravian metabolism was not at the levels seen in extant birds, it was sufficient to perform short burst activities (Erickson et al., 2009). Regardless, as our methodology uses wing-beat frequency in conjunction with body size and wing arc measures to generate a lift production value, we are not dependent on either theory (power or lift force) to produce meaningful results.

Taxonomic sampling

Forty-five specimens representing twenty-four non-avian theropod taxa and five avian taxa were examined. Non-avian theropod specimens ranged in mass from approximately 60 g to 18 kg (Tables 1 and S1). Of these, twenty-eight are from specimens accounting for twelve non-avian theropod taxa with preserved feather material, the rest are from closely related taxa that are inferred to be feathered and were included to broaden the scope of the maniraptorans represented. We a priori excluded the tyrannosaurids Yutyrannus, because of its large size (estimated mass ∼1,400 kg), and Dilong, due to its incompletely preserved forelimb. Multiple individuals were included for Anchiornis, Similicaudipteryx, Caudipteryx, Microraptor, Sinosauropteryx, Mei, Archaeopteryx, Jeholornis, and Sapeornis to represent different size classes and ontogenetic stages as different stages in ontogeny may have different life history strategies (Parsons & Parsons, 2015). To address the possibility of WAIR in juvenile but not adult members of Pennaraptora, three late stage embryos: MOR 246-1 Troodon formosus per Varricchio, Horner & Jackson (2002), MPC-D100/971.

Table 1 Fossil taxa examined in this study.

Taxa in bold were specimens without preserved forelimb remegies for whom feather lengths were estimated based on closely related taxa or other members of the same genus. For Jianchangosaurus we based our estimate on the longest preserved body feather traces, this is defensible as this clade is not know to have pennaceous remegies (Foth, Tischlinger & Rauhut, 2014) and in other maniraptorans without remegies the integument on the distal cervicals are similar in size, if not longer, than those on the forelimbs (Currie & Chen, 2001). CF indicates mass estimated based on Christiansen & Fariña (2004), Liu indicates avian mass estimates based on Liu, Zhou & Zhang (2012), Fe for avian mass estimates based on Field et al. (2013). See text for discussion of body mass calculations and wing beat frequencies.

Taxa	Reference	Wing length (m)	Span (m)	Mass (kg) CF	Mass (kg) Liu	Mass (kg) FE	Wing Area (m^2)	Wing loading N/M2	
Anchiornis	Li et al. (2010)	0.16	0.33	0.09	–	–	0.01	70	
Anchiornis	Sullivan et al. (2010)	0.24	0.50	0.38	–	–	0.03	146	
Archaeopteryx	Foth, Tischlinger & Rauhut (2014)	0.31	0.65	–	0.24	–	0.06	38	
Archaeopteryx	Foth, Tischlinger & Rauhut (2014)	0.31	0.65	–	–	0.36	0.06	57	
Archaeopteryx	Mayr et al. (2007)	0.29	0.61	–	0.23	–	0.06	38	
Archaeopteryx	Mayr et al. (2007)	0.29	0.61	–	–	0.32	0.06	55	
Archaeopteryx	Elżanowski (2002)	0.33	0.69	–	0.31	–	0.07	45	
Archaeopteryx	Elżanowski (2002)	0.33	0.69	–	–	0.48	0.07	70	
Archaeopteryx	Mayr et al. (2007), Nudds & Dyke (2010)	0.26	0.55	–	0.18	–	0.05	38	
Archaeopteryx	Mayr et al. (2007), Nudds & Dyke (2010)	0.26	0.55	–	–	0.25	0.05	53	
Archaeopteryx	Mayr et al. (2007)	0.27	0.57	–	0.19	–	0.05	36	
Archaeopteryx	Mayr et al. (2007)	0.27	0.57	–	–	0.27	0.05	51	
Archaeopteryx@	Mayr et al. (2007)	0.19	0.39	–	0.11	–	0.02	47	
Archaeopteryx@	Mayr et al. (2007)	0.19	0.39	–	–	0.14	0.02	60	
Aurornis*	Godefroit et al. (2013)	0.22	0.47	0.38	–	–	0.02	160	
Caudipteryx	Zhou & Wang (2000)	0.35	0.72	5.52	–	–	0.09	631	
Caudipteryx	Sullivan et al. (2010)	0.28	0.58	3.77	–	–	0.04	863	
Changyuraptor#	Han et al. (2014)	0.68	1.42	5.64	–	–	0.43	130	
Citipati MPC-D100/971	Lü et al. (2013)	0.11	0.22	0.05			0.00	397	
Confuciusornis	Chiappe et al. (1999)	0.32	0.67	–	0.14	–	0.09	15	
Confuciusornis	Chiappe et al. (1999)	0.32	0.67	–	–	0.19	0.09	20	
Eoconfuciusornis	Sullivan et al. (2010)	0.22	0.46	–	0.09	–	0.04	24	
Eoconfuciusornis	Sullivan et al. (2010)	0.22	0.46	–	–	0.12	0.04	30	
Eosinopteryx	Godefroit et al. (2013)	0.16	0.33	0.14	–	–	0.01	111	
Jeholornis	Ji et al. (2002)	0.41	0.86	–	0.34	–	0.12	29	
Jeholornis	Ji et al. (2002)	0.41	0.86	–	–	0.54	0.12	45	
Jeholornis*	Zhou & Zhang (2002)	0.55	1.15	–	0.60	–	0.21	28	
Jeholornis*	Zhou & Zhang (2002)	0.55	1.15	–	–	1.05	0.21	49	
Jianchangosaurus	Pu et al. (2013)	0.40	0.83	14.70	–	–	0.03	5,018	
Jinfengopteryx*	Ji et al. (2005)	0.17	0.37	0.46	–	–	0.01	317	
Mahakala#	Turner, Pol & Norell (2011)	0.20	0.42	0.67	–	–	0.03	229	
Mei long*	Gao et al. (2012)	0.12	0.26	0.36	–	–	0.01	505	
Mei long*	Xu & Norell (2004)	0.15	0.31	0.73	–	–	0.01	714	
Microraptor	Li et al. (2012)	0.24	0.50	0.17	–	–	0.04	46	
Microraptor	Xu et al. (2003), Sullivan et al. (2010)	0.41	0.86	0.88	–	–	0.12	69	
Microraptor hanqingi#	Gong et al. (2012)	0.47	0.98	2.05	–	–	0.18	110	
Oviraptor incertae sedis MPC-D100/1018	Lü et al. (2013)	0.09	0.19	0.03			0.00	305	
Protarchaeopteryx	Ji & Ji (1997)	0.26	0.54	2.58	–	–	0.02	1,445	
Sapeornis	Pu et al. (2013)	0.44	0.92	–	0.51	–	0.12	43	
Sapeornis	Pu et al. (2013)	0.44	0.92	–	–	0.88	0.12	74	
Sapeornis*	Zhou & Zhang (2003a) and Zhou & Zhang (2003b)	0.57	1.21	–	0.80	–	0.20	40	
Sapeornis*	Zhou & Zhang (2003a) and Zhou & Zhang (2003b)	0.57	1.21	–	–	1.49	0.20	74	
Similicaudipteryx	Xu et al. (2009), Dececchi & Larsson (2013)	0.40	0.84	4.23	–	–	0.12	345	
Similicaudipteryx	Xu et al. (2009), Dececchi & Larsson (2013)	0.07	0.15	0.06	–	–	0.00	372	
Sinocalliopteryx	Sullivan et al. (2010)	0.37	0.77	18.43	–	–	0.05	3,596	
Sinornithoides	Russell & Dong (1993)	0.31	0.77	18.4	–	–	0.04	1,151	
Sinornithosaurus	Ji et al. (2001)	0.26	0.54	1.94	–	–	0.02	1,032	
Sinornithosaurus	Sullivan et al. (2010)	0.19	0.41	0.29	–	–	0.01	229	
Sinosauropteryx	Currie & Chen (2001)	0.10	0.20	0.88	–	–	0.00	4,755	
Sinosauropteryx	Currie & Chen (2001)	0.05	0.09	0.19	–	–	0.00	11,910	
Sinovenator*	Benson & Choiniere (2013)	0.24	0.50	2.44	–	–	0.03	919	
Tianyuraptor	Chan, Dyke & Benton (2013), Dececchi & Larsson (2013)	0.39	0.82	13.36	–	–	0.06	2,272	
Troodon Embryo MOR 246-1	Varricchio, Horner & Jackson (2002)	0.08	0.16	0.05			0.00	214	
Xiaotingia*	Xu et al. (2011)	0.24	0.50	0.82	–	–	0.03	305	
Yixianosaurus	Dececchi, Larsson & Hone (2012)	0.29	0.61	1.30	–	–	0.04	323	
Yixianosaurus	Dececchi, Larsson & Hone (2012)	0.29	0.61	1.89	–	–	0.04	470	
Yulong%$	Lü et al. (2013)	0.18	0.38	0.50	–	–	0.02	280	
Zhenyuanlong	Lü & Brusatte (2015)	0.58	1.22	11.99	–	–	0.23	515	
Notes:

@ Based on other Archaeopteryx specimens.

# Denotes estimates based on Microraptor gui.

* Based on Anchiornis.

$ Based on Caudipteryx.

Citipati osmolskea and MPC-D100/1018 Oviraptor incertae sedis per Lü et al. (2013) were included in this analysis. These specimens are incomplete, but forelimb lengths could be estimated based on the fact that the humerus/forelimb ratio in non-avian and basal avian theropods does not change significantly across ontogeny (Table S2). We used the value of ∼43% MOR 246-1 based on the ratios seen in other Troodontids (range between 39–45%) based on Mei, Jinfengopteryx, Anchiornis, Aurornis, Sinovenator, Sinornithoides and Xiaotingia. For MPC-D100/971 and MPC-D100/1018 we used 41% based on Citipati. For all late stage embryos we reconstructed wing area as if they possessed wings with pennaceous feathering proportional to that seen in adults. This is likely an overestimation, as hatchling and young juveniles in other non-avian theropods do not show pennaceous development to the extent of adults (Xu et al., 2009, Zelenitsky et al., 2012).

Mass estimations for non-avian theropods were based on values for femur length (Christiansen & Fariña, 2004) except for Yixianosaurus, which has no preserved hindlimbs, for whom upper and lower mass estimate boundaries were taken from Dececchi, Larsson & Hone (2012). As non-avian and avian theropods show significant difference in hindlimb scaling (Dececchi & Larsson, 2013), this method could not be applied to the avian theropods in our dataset. For birds, two mass estimates were generated from the regressions derived from humerus length equations of extant birds (Liu, Zhou & Zhang, 2012; Field et al., 2013), this element was selected as it showed high correlation values in both source datasets and were easily computable for all specimens. Nodal values were calculated based on a modified version of the phylogeny in Dececchi & Larsson (2013) (Data S1).

Wing dimensions

Wing length was calculated based on the length of the humerus, ulna, metacarpal II, and the longest primary feather length, arranged in a straight line. Metacarpal length was used instead of total manus length as the longest primaries attach to the metacarpals and distal forelimb in paravians (Savile, 1957; Elżanowski, 2002; Xu, Ma & Hu, 2010; Foth, Tischlinger & Rauhut, 2014). This gives values similar to those previously reported for maximal straight-line length of the wing in Archaeopteryx, differing by less than 1% (Yalden, 1971). Wing area was estimated using a chord value 65% of maximum primary length based on the differences between the longest primary feather and the shortest, distal primary in Archaeopteryx (Elżanowski, 2002; Foth, Tischlinger & Rauhut, 2014) and Caudipteryx (Qiang et al., 1998). This estimate produces a greater wing area, by 15%, than what was calculated by Yalden (1971) for the Berlin specimen of Archaeopteryx and produces similar overestimations for other paravian taxa with published wing areas such as Microraptor (+38% compared to Chatterjee & Templin (2007) estimate and +9% over that of Alexander et al. (2010) and Zhenyuanlong (5% greater than calculated by Lü & Brusatte (2015)). Therefore, we treat our values as upper bound estimates of maximum wing area as they are also overestimates of functional wing area since they ignore the natural flexed position that the limbs take during locomotion. We used this value for our primary analysis as it gives highest possible values for all our force production data and thus the maximum likelihood of success in achieving the minimum threshold values indicating the possible presence of a behavior in said taxon. For taxa without primary feathers preserved (Table 1), we estimated their length based on either other members of the same genus or closely related taxa and assuming congruent lengths. We estimated body width using furcular widths (Table S3) this represents an addition of between 10–15% to the value of the non-avian theropod skeletal arm span. In extant bird wings feathers add another 40 + % to skeletal arm length (Nudds, Dyke & Rayner, 2007) and proportionally more in many non-avian theropods (Table 1). Wingspan was set 2.1 times wing length (feather lengths included) to assure we did not underestimate the potential wingspan and the influence of the body on wing area in non-avian taxa.

Model construction

To test WAIR, flap running, and vertical leaping we used equations based on those of Burgers & Chiappe (1999) and on extant bird flight work in Pennycuick (2008) to estimate force production in a similar context to what is examined here.

bw=0.5Cl*p*(fAmp+U)2S/9.8*M

Where bw denotes the proportion of body weight supported by the lift generated by the wings (see Supplemental Materials Section S4 for more complete description of all formula and calculations). This relatively simple model was chosen as it is easier to update with new paleobiological information and allowed us to see directly the result of varying the input data to see how varying models of theropod functional limitations shape the results. To test the accuracy of our model, we compared our body weight support results to published data for Chukar partridges during WAIR across the three ontogenetic stages, Pigeon data during WAIR, and birds during takeoff (Table 2). Our values are within the range seen in published data for all three stages of WAIR development and show values greater than 1.0 for all birds undertaking leaping takeoff. As our simple model accurately matches real world experimentally derived values of extant taxa, we believe it a suitable starting point to derive comparative force production data for fossil avian and non-avian theropods.

Table 2 Results of equations for calculating forces produced during WAIR and takeoff using data from extant avians.

For Chukars body mass, wing area and body velocity are based on Tobalske & Dial (2007), Flapping frequency and angle are based on Jackson, Segre & Dial (2009). Coefficient of lift values (Cl) based on Heers, Tobalske & Dial (2011). For pigeons WAIR all data based on Jackson, Tobalske & Dial (2011) except for wing area, which is taken from pigeons Crandell & Tobalske (2011) from pigeons with similar mass and wing length. For avian takeoff values are based on Tobalske & Dial (2000) and Askew, Marsh & Ellington (2001).

Taxon	Stage	Body Mass (kg)	Wing Area (m^2)	Flap angle (rad)	Wing beat (Hz)	Velocity (m/s)	BW	BW	BW	BW	
		Cl = 1.0	Cl = 1.2	Cl = 1.5	Cl = 1.6	
Chukar	I	0.024	0.0036	1.57	22	0.60	0.06	0.08	–	–	
Chukar	II	0.222	0.0297	2.5	18.7	1.20	0.85	1.02	–	–	
Chukar	III	0.605	0.0499	2.16	18.7	1.50	0.65	0.78	0.97	1.02	
Pigeon	WAIR 65°	0.42–0.47	0.067	1.57	6.2–6.7	1.50	0.21–0.26	0.25–0.31	0.31–0.39	0.33–0.41	
Pigeon	WAIR 85°	0.42–0.47	0.067	1.57	7.3–7.7	1.50	0.28–0.31	0.34–0.37	0.42–0.46	0.45–0.49	
Northern bobwhite	Take off	0.199	0.0243	2.44	19.9	3.25	–	–	–	1.25	
Chukar	Take off	0.4915	0.0483	2.64	16.1	2.87	–	–	–	1.62	
Ring necked pheasant	Take off	0.9434	0.1002	2.64	11	2.34	–	–	–	1.37	
Turkey	Take off	5.275	0.3453	2.79	7.6	2.32	–	–	–	1.26	
Blue breasted quail	Take off	0.0436	0.0098	2.44	23.2	4.81	–	–	–	2.42	
Harris hawk	Take off	0.92	0.119	2.60	5.8	4.13	–	–	–	2.07	
Pigeon	Take off	0.307	0.0352	2.48	9.1	2.62	–	–	–	1.19	

Creation of benchmarks

As WAIR ability is not uniform across ontogeny and seems to be linked to force production (Jackson, Segre & Dial, 2009), we created two-benchmarks of proportion of body mass supported for taxa to reach. Values between 0.06–0.49 body weight (bw) are classified as level 1 WAIR, which corresponds to the earliest stages of ontogeny and sub vertical ascents (late stage I and early stage II per Jackson, Segre & Dial, 2009) with greater than 50% contribution to external vertical work generated by the hindlimbs (Bundle & Dial, 2003). 0.5 bw and greater denote level 2 WAIR, equivalent to more mature Stage II and III individuals (per Jackson, Segre & Dial, 2009) which are capable of high angle to vertical ascents and whose forelimbs become more prominent in force production (Bundle & Dial, 2003). Although we understand the transition between stages during WAIR is semi-artificial, we wished to create a classification scheme that corresponds to the different levels of WAIR capabilities seen in extant systems (Jackson, Segre & Dial, 2009). The selection of 0.06 bw for achieving stage I was chosen to represent real world recorded minima for this behavior and thus should be considered minimal levels achieved before reconstructions of WAIR are accepted.

Coefficient of lift (Specific lift)

We examined potential performance during the wing-driven phase of flap-running, WAIR, and leaping takeoff in our analyses. As a result, all three of the behaviors are subject to constraints of lift production efficiency. The production of lift relative to planform area, speed, and fluid density is summarized as the coefficient of lift.

During WAIR analysis, a coefficient of lift (CL) of 1.0 was used. This corresponds to a value estimated during WAIR use in juvenile Chukars at early stage II (10 dph) (Heers, Dial & Tobalske, 2014) but greater than that in the earlier ontogenetic stages (Heers, Tobalske & Dial, 2011). We selected this value as this age class has been proposed to be analogous to derived maniraptoran theropod capabilities such as Anchiornis and Microraptor and this Cl is achievable by all ontogenetic stages beyond 5 dph depending on the angle of attack (Heers, Dial & Tobalske, 2014). For leaping takeoff we used a Cl of 1.5, which corresponds to the minimal values estimated in adult Chukars during high angle WAIR (Heers, Tobalske & Dial, 2011) and below the 1.64 calculated for the pigeon during takeoff (Usherwood, 2009). For flap running, we used the equations of Burgers & Chiappe (1999) with the following modifications: we ran permutations for all three downstroke (50, 70 and 90°) angles not just 50° as per the original analysis and reduced the Cl to 1.2 from 2. We choose to make the Cl closer to that estimated during late stage Chukar WAIR attempts (Heers, Tobalske & Dial, 2011) as WAIR is simply a specific use case of flap running on a highly angled substrate. This value is achievable by Chukars older than 20 dph (Heers, Dial & Tobalske, 2014). Using the Cl of non-volant and juvenile Chukar both produces reasonable minimum values for these behaviours and more closely simulates the expected outputs in non-avian theropods before powered flight.

During low advance ratio wing-driven behaviors (launch, landing, WAIR, etc.), the coefficient of drag can be quite large. In young Chukars, the coefficient of drag can be near the coefficient of lift, thereby potentially providing a significant component of weight support during controlled descent or significantly affecting reaction forces during WAIR (Heers, Tobalske & Dial, 2011). To confirm that using pure Cl as our specific fluid force coefficient was an accurate approach (instead of the total fluid resultant with both Cl and Cd), we compared predicted reaction forces and weight support to values measured in vivo and reported in the literature (Tobalske & Dial, 2007; Heers, Dial & Tobalske, 2014). Because a close match was found across multiple size classes, we assume for the remainder of the calculations that reaction forces during WAIR are not greatly affected by a high coefficient of drag (though we note that for controlled descent or burst climb out, behaviors we did not investigate, high Cd is likely a critical component).

Wing beat frequency

Wing beat frequencies scale negatively to body mass in steady flight (Greenewalt, 1975; Pennycuick, 2008) and takeoff (Askew, Marsh & Ellington, 2001; Jackson, 2009) across species in extant birds. Wingbeat frequencies during takeoff are similar to those during WAIR (Tobalske & Dial, 2007). For this study we used the maximum takeoff wingbeat frequency regressions from Jackson (2009) for all birds in his sample (see all Supplemental Tables), and for only ground foraging birds (GF), we also added Galliformes takeoff data from Askew, Marsh & Ellington (2001) to Jackson’s dataset to produce a third regression equation (MOD). For the MOD dataset we incorporated a phylogenetic correction using PDAP v 1.15 (Midford, Garland & Maddison, 2010), with branch lengths based on divergence times derived from the chronograms of Jetz et al. (2012) (Data S2).

Wing range of motion

Abduction of the forelimb beyond the horizontal plane that transects the vertebral column was not possible in most non-avian theropods resulting in a maximum stroke angle for forelimb motion to be less than 90° (Senter, 2006a; Senter, 2006b; Turner, Makovicky & Norell, 2012). The glenoid fossa faces ventrolaterally in these taxa and only shifted to a more lateral configuration at Paraves (Makovicky & Zanno, 2011; Turner, Makovicky & Norell, 2012). The glenoid continued to translate upward until reaching the dorsolaterally facing position of most extant birds at the phylogenetic level of Jeholornis and Sapeornis (Zhou & Zhang, 2003a; Zhou & Zhang, 2003b).

Extant birds have extensive shoulder abductive ranges. For example, during WAIR, the abductive flap angle of juvenile Chukars ranges from 90° at stage I to greater than 143° at stage II (Jackson, Segre & Dial, 2009). Images show that in all cases, the forelimb ascends to a vertical or slightly beyond position (see Tobalske & Dial, 2007, Figs. 4 and 6; Jackson, Segre & Dial, 2009, Fig. 1; Heers, Dial & Tobalske, 2014, Fig. 1).

Given the abduction limitations of the non-avian theropod glenoid, we chose flap angles of 50, 70 and 90° to encapsulate the range of values expected across Theropoda and ran them for all taxa. An angle of 90° is likely unattainable for all non-avian theropods due to the constraints of reducing contact with the substrate on the latter part of the downstroke and shoulder morphology since the humerus cannot exceed the dorsal rim of the glenoid which is aligned with the vertebral axis (or vertebral frame of reference per Dial, Jackson & Segre, 2008). It was included to create an upper bracket on possible support values.

Velocities for the center of mass used for the different analyses were based on those of extant birds. For WAIR used as our assigned velocity 1.5 m/s based on the speed of adult birds (Tobalske & Dial, 2007). This is higher than achieved for the early, pre-flight capable ontogenetic stages (0.6 m/s in stage I, 1.2 m/s in stage II), and thus acts as a fair upper velocity bound, though it is likely beyond the capabilities of non-avian theropods with less developed wings. For leaping we calculated three values: height gain if wing thrust was added to that generated by the hindlimbs, vertical distance increase given the increased take off velocity due to flapping and takeoff potential from a standing jump. Calculating height and distance gain was done through a modification of existing equations used to model pterosaur launch (Witton & Habib, 2010) to account for the bipedal nature of non-avian theropods (see Supplementary Information for these equations). To compensate for the effects of body size, a scalar is introduced to ensure the pre-loading values would be 2.4, a conservative value well within the range seen in extant tetrapods (Biewener, 2003). Our pre-loading scalar accounts for the fact that animals gain significant power amplification from the release of stored elastic energy in their limbs. Even in non-specialist jumpers this amplification can be greater than twice the maximum mass specific power derived from the muscles and in specialist can be 10 × higher or more (Henery, Ellerby & Marsh, 2005 and references therein). For leaping takeoff, our starting inputs were two different takeoff speeds recorded in on extant avians (Earls, 2000; Tobalske & Dial, 2000; Askew, Marsh & Ellington, 2001). Higher values for leaping have been recorded in some mammals (Günther et al., 1991) and after several wing beats in birds (Askew, Marsh & Ellington, 2001; Berg & Biewener, 2010), thus these values may not represent the maximal possible values for small theropods. For flap running the assigned start value was 2 m/s, which is the same starting velocity used in Burgers & Chiappe (1999). This speed is well within the range of sprint speeds of many lizards (Huey, 1982; Christian & Garland, 1996; Irschick & Jayne, 1999) and small mammals (Iriarte-Díaz, 2002), whereas many terrestrial birds can sustain this speed for over thirty minutes (Gatesy & Biewener, 1991; Gatesy, 1999). These values are likely well below the maximum sprint speed of these taxa (Sellers & Manning, 2007) but allowed us to determine if there was significant increase in speed using the wing generated thrust alone.

We excluded the potential drag and damage caused by hindlimb feathers of some paravians through contact with the substrate. At low hindlimb angles used during the ascent of inclined surfaces (see the metatarsus during WAIR in Fig. 1 from Jackson, Segre & Dial, 2009) the distal limb feathers would have contacted the surface and caused frictional drag, which would have reduced performance and damaged the feathers (Dececchi & Larsson, 2011). Although these variables may have evolved throughout the transition from theropods into early birds, treating them as constants provided a “best case scenario” for non-avian theropods constraining the upper limits for when these behaviours were possible.

Figure 1 Wing loading values in non-avian theropods.

Each open circle denotes the value per specimen for taxa with multiple specimens included in analysis. Note that only a minority of paravian specimens are below the lines denoting values pre WAIR quadruped crawling in Chukar (3 dph) and when fledging occurs (10 dph) as well as WAIR capable Brush Turkeys.

Wing contribution to leaping

Three additional estimates for wing contributions to vertical leaping were made. The first estimates the percentage increase possible to the maximum leap through the addition of thrust generated by flapping. This calculation assumed the maximum wing output occurred at the top of the leap arch, and that the forces generated were directed vertically. This was done through a modification of the terrestrial launch methodology of Witton & Habib (2010, see Data S3) to accommodate bipedal theropod models with and without wing generated thrust. The difference between the maximum heights gained with wing generated thrust was presented as a percentage increase (see Datas S3 and S4 for more detailed description of the equations used and a sample calculation spreadsheet). The second evaluates the horizontal distance extension to a leap through the addition of flapping generated thrust. This was calculated by using the speed at takeoff generated by the equations for bipedal launch (see Datas S3 and S4) at both 30 and 45° launch angle. The later corresponds to the theoretical best angle for a projectile while the former more closely resembles the angle of takeoff measured in human and lizard leapers (Toro, Herrel & Irschick, 2004; Linthorne, Guzman & Bridgett, 2005; Wakai & Linthorne, 2005). In both cases our models were treated as if there was no difference in takeoff and landing height, thus making the calculation of jump distance Djump=(v2sin2Θ)/g

Where v equals the takeoff velocity and Θ the angle of takeoff.

Vertical take offs were deemed possible when body weight (bw) support values were equal to or greater than 1.0 using the speed and lift parameters mentioned above.

Results

Wing loading

Increase in WAIR ability broadly corresponds to decreased wing loading in Chukars (Heers & Dial, 2015), something noted in other galliform birds (Dial & Jackson, 2011). Thus wing loading values may offer a rough comparison between non-avian theropod specimens and Chukars of a similar body mass. Among non-avian theropods, wing loading values ranged from 46 N/m2 (Microraptor) to over 11,000 N/m2 (Sinosauropteryx). Of the thirty-four non-avian specimens included, only eight, representing five genera (all are deinonychosaurs) showed loading values less than that seen in 1-day-old Chukars (170 N/m2), the highest values recorded across ontogeny. 1-day-old Chukar chicks do not WAIR, can only surmount inclines of less than 48° still performed asynchronous wing beats and their wings make prolonged contacts with the substrate in a crawling fashion (Jackson, Segre & Dial, 2009; Heers & Dial, 2015). No non-paravian showed values less than the 160 N/m2 measured at 3 dph Chukars, with most pennaraptorans at values 2–8 times that seen at even the highest Chukar chick loadings (Table 1; Fig. 1). Focusing on the embryonic and early ontogenetic stage specimens in our analysis, to test whether WAIR was possible at early ages and lost through ontogeny, we recovered loading values again significantly higher than the highest values seen during Chukar ontogeny, with values 126–234% those of 1-day-old chicks which were also significantly smaller. For comparison, the hatchling size Similicaudipteryx specimen (STM 4-1) had a body mass estimated at approximately 63 g, similar to a 17 dph Chukar chick (stage II), but wing loading values of 372 N/m2, 5.8 times higher than seen in the 17 dph chick and over twice that seen in 3 dph Chukars due to Similicaudipteryx having a wing area only the size of a 6 dph chick which weight approximately 16 g. This suggests that none of the non-paravian theropods could perform the lowest levels of WAIR, even disregarding their limited range of motion and flapping frequency compared to juvenile extant avians. None of the Mesozoic avian taxa, under either mass reconstruction, showed loading values above 74 N/m2, which corresponds to approximately 11 dph (stage II) Chukar chicks, which is approximately the time where fledgling begins (Harper, Harry & Bailey, 1958; Christensen, 1996).

WAIR

At a CoM velocity of 1.5 m/s nine of thirty-four specimens of non-avian theropods reached the minimal benchmark for level 1 WAIR (0.06 bw) under at least one of the three flapping speed and flap angle permutations (Fig. 2; Tables 3 and S4–S6). When the velocity was decreased to 0.6 m/s number that succeed decreased to eight as the Sinornithosaurus specimen based on the measurements of Sullivan et al. (2010) failed to achieve the 0.06 bw benchmark (Fig. 2; Table 3). All are deinonychosaurs. Three specimens (the larger Similicaudipteryx specimen, and the smaller mass estimates for Yixianosaurus and Yulong) approach the WAIR level 1 criteria, but none yield values higher than 0.05 bw, and this only under the MOD reconstruction at the highest abduction angle. All specimens of Microraptor and the smaller specimens of Anchiornis and Eosinopteryx yielded bodyweight support values above 0.06 bw across all permutations at 1.5 m/s whereas at 0.6 m/s only the smaller Anchiornis and Microraptor gui specimens achieve this. Within non-avian theropods using a 90° flap angle at 1.5 m/s, only a single specimen of Microraptor gui (BMNHC PH881) has body weight support values reaching the 0.5 bw cutoffs for WAIR level 2, though the larger specimen (IVPP V 13352) comes close under the MOD reconstruction (Tables 3 and S4–S6). At 50° only the smaller Anchiornis, Changyuraptor, Eosinopteryx and all 3 Microraptor specimens, achieve the 0.06 bw benchmark at 1.5 m/s and this decreases to only the smaller Anchiornis and Microraptor at 0.6 m/s. No non-avians or Archaeopteryx achieved bw support values higher than 0.33 under the 50° at 1.5 m/s and only Microraptor gui, Archaeopteryx specimens and the smaller Anchiornis reaching a minimal of 0.1 bw under this permutation.

Figure 2 Evolution of WAIR performance.

Estimated evolutionary ranges of WAIR stages I and II (Dial, 2003; Heers & Dial, 2012; Heers, Dial & Tobalske, 2014) are mapped over a phylogeny of selected Maniraptoriformes. Upper lines are for 90° flap angles and lower lines for 50° flap angles. Flight-stroke specific characters are mapped onto the phylogeny: 1, forelimb integument; 2, pennaceous feathers on forelimb; 3, L-shaped scapulocoracoid; 4, laterally facing glenoid; 5, asymmetrical remigies; 6, alula; 7, incipient ligament-based shoulder stabilization; 8, dorsolaterally facing glenoid; 9, full ligament-based shoulder stabilization. The bottom coloured lines denote 50° flap angles and upper coloured lines 90°. Silhouettes from PhyloPic images by B. McFeeters, T.M. Keesey, M. Martynuick, and original.

Table 3 Table of body wight support values across specimens under 90° flap angle.

Body weight (bw) support values across non-avian and basal avian taxa under three different flapping frequency estimators (see text for description). Calculations are based on the 90° flap angle permutation at two velocity of the centre of mass (0.6 and 1.5 m/s). This correspond to recorded velocity of earliest WAIR capable juveniles (0.6 m/s) and adult (1.5 m/s) Chukars (Tobalske & Dial, 2007).

Taxa	Specimen	M/S	bw All	bw GF	bw MOD	M/S	bw All	bw GF	bw MOD	
Anchiornis	BMNHCPH828	1.5	0.24	0.22	0.22	0.6	0.17	0.15	0.15	
Anchiornis	LPM B00169	1.5	0.10	0.09	0.12	0.6	0.06	0.06	0.08	
Archaeopteryx	11th	1.5	0.70/0.37	0.62/0.33	0.78/0.46	0.6	0.52/0.27	0.45/0.23	0.59/0.34	
Archaeopteryx	Berlin	1.5	0.67/0.38	0.60/0.34	0.74/0.46	0.6	0.50/0.27	0.43/0.24	0.56/0.34	
Archaeopteryx	London	1.5	0.57/0.28	0.50/0.25	0.67/0.37	0.6	0.42/0.20	0.37/0.17	0.51/0.27	
Archaeopteryx	Munich	1.5	0.66/0.39	0.59/0.34	0.68/0.43	0.6	0.48/0.28	0.42/0.24	0.51/0.32	
Archaeopteryx	Thermopolis	1.5	0.71/0.41	0.63/0.37	0.75/0.47	0.6	0.52/0.29	0.46/0.26	0.56/0.34	
Archaeopteryx	Eichstatt	1.5	0.42/0.29	0.38/0.26	0.39/0.28	0.6	0.30/0.20	0.26/0.17	0.27/0.19	
Aurornis	YFGP-T5198	1.5	0.08	0.07	0.10	0.6	0.05	0.05	0.07	
Caudipteryx	IVPP 12344	1.5	0.01	0.01	0.02	0.6	0.01	0.00	0.01	
Caudipteryx	IVPP 12430	1.5	0.01	0.01	0.01	0.6	0.00	0.00	0.01	
Changyuraptor	HG B016	1.5	0.11	0.10	0.25	0.6	0.05	0.05	0.14	
Citipati	MPC-D100/971	1.5	0.03	0.03	0.03	0.6	0.02	0.02	0.02	
Eosinopteryx	YFGP-T5197	1.5	0.12	0.11	0.12	0.6	0.08	0.07	0.08	
Jianchangosaurus	41HIII-0308A	1.5	0.00	0.00	0.00	0.6	0.00	0.00	0.00	
Jinfengopteryx	CAGS-IG 04-0801	1.5	0.03	0.02	0.03	0.6	0.02	0.01	0.02	
Mahakala	IGM 100/1033	1.5	0.04	0.03	0.05	0.6	0.02	0.02	0.03	
Mei long	DNHM D2154	1.5	0.01	0.01	0.02	0.6	0.01	0.01	0.01	
Mei long	IVPP V12733	1.5	0.01	0.01	0.01	0.6	0.00	0.00	0.01	
Microraptor	BMNHC PH 881	1.5	0.49	0.43	0.50	0.6	0.35	0.31	0.36	
Microraptor	IVPP V 13352	1.5	0.28	0.25	0.42	0.6	0.20	0.17	0.32	
Microraptor hanqingi	LVH 0026	1.5	0.14	0.12	0.24	0.6	0.08	0.07	0.15	
Oviraptor in sedis	MPC-D100/1018	1.5	0.05	0.04	0.03	0.6	0.03	0.03	0.02	
Protarchaeopteryx	GMV2125	1.5	0.00	0.00	0.01	0.6	0.00	0.00	0.00	
Similicaudipteryx	STM22-6	1.5	0.02	0.02	0.05	0.6	0.01	0.01	0.03	
Similicaudipteryx	STM4-1	1.5	0.02	0.02	0.02	0.6	0.01	0.01	0.01	
Sinocalliopteryx	JMP-V-05-8-01	1.5	0.00	0.00	0.00	0.6	0.00	0.00	0.00	
Sinornithoides	IVPP V9612	1.5	0.01	0.01	0.01	0.6	0.00	0.00	0.01	
Sinornithosaurus	NGMC-91A	1.5	0.01	0.01	0.01	0.6	0.00	0.00	0.01	
Sinornithosaurus	Sullivan et al. (2010)	1.5	0.05	0.05	0.06	0.6	0.03	0.03	0.04	
Sinosauropteryx	NICP 127587	1.5	0.00	0.00	0.00	0.6	0.00	0.00	0.00	
Sinosauropteryx	NIGP 127586	1.5	0.00	0.00	0.00	0.6	0.00	0.00	0.00	
Sinovenator	IVPP V11977	1.5	0.01	0.01	0.01	0.6	0.00	0.00	0.01	
Tianyuraptor	STM1–3	1.5	0.00	0.00	0.00	0.6	0.00	0.00	0.00	
Troodon embryo	MOR 246-1	1.5	0.04	0.04	0.03	0.6	0.02	0.02	0.02	
Xiaotingia	STM 27-2	1.5	0.03	0.03	0.05	0.6	0.02	0.02	0.03	
Yixianosaurus	IVPP 12638	1.5	0.03/0.02	0.03/0.02	0.05/0.03	0.6	0.02/0.01	0.02/0.01	0.03/0.02	
Yulong	41HIII-0107	1.5	0.03	0.03	0.04	0.6	0.02	0.02	0.02	
Zhenyuanlong	JPM-0008	1.5	0.02	0.01	0.04	0.6	0.01	0.01	0.03	

Among Mesozoic birds, the different mass estimation methods produced significantly different body weight support values and are more prominent in the most basal birds in our analysis Sapeornis and Jeholornis (Fig. 2; Tables S4–S6). All basal avians show the capability of level 1 WAIR (bw support values of 0.06 or greater) under all flap frequencies estimates, mass estimates or flap angles used here and no avians showing values below 0.1 bw under any permutation. In Archaeopteryx, there is no clear trend in WAIR capability and allometry as all specimens besides the Eichstatt individual show a similar range of body weight support values (Table 3). At the higher flap angle and lower mass, all avians show the capability for level 2 WAIR (> 0.5 bw). All birds more derived than Archaeopteryx yield a body weight support values in excess of 1.0 bw at their lower mass estimate at 1.5 m/s 90° flap angle under all 3 flap frequencies, except for Sapeornis where the smaller specimen exceeds 1.0 bw only under the MOD permutation. Of note, the values recovered for more derived avians are significantly higher than those observed in experimental data (Tobalske & Dial, 2007) or calculated using extant measurements (Tables 2 and S7) and well above the 1.0 threshold for takeoff. This suggests that these taxa could have performed this behavior at lower wing beat frequencies, body velocities and flap angles than the values used here, as seen in some extant birds (Jackson, Tobalske & Dial, 2011), or that physiology and power production differed between extant and basal birds (Erickson et al., 2009; O’Connor & Zhou, 2015), or a combination of both. If the latter is correct, it suggests our measurements for non-avian theropods overestimate the power production potential in these taxa, and thus overestimate their WAIR capabilities.

Flap running

Among non-avian theropods, flap running peaked in effectiveness within small-bodied paravians (Fig. 3; Table S8). With a 90° flap angle, the smaller Anchiornis specimen and Microraptor gui were the only non-avian taxa to show increases greater than 1.0 m/s under all permutations (71–79 and 75–208% performance increases, respectively), although only Microraptor achieved speeds capable of flight. More realistic 50° flap angles yielded only a 23–27 and 26–65% performance increase for these taxa. Among non-paravians, even under the highest flap angle and flap frequency permutations no taxon exceeded an increase of 17% in running speed with the highest values found in the larger specimen of Similicaudipteryx. At flap angles below 90° only the larger Similicaudipteryx and the lighter mass estimated Yixianosaurus specimens among non-paravians yielded velocity increases approaching 10%. Although some paravians had high levels of increased speed, Mahakala, Mei, Jinfengopteryx, Xiaotingia, Tianyuraptor, and Sinovenator showed increases of less 17% under all permutations, with many showing values in the single digits. At 50° only Microraptor sp., Changyuraptor, Eosinopteryx and Anchiornis showed a greater than 10% increase in running velocity. All specimens of Archaeopteryx showed speed increases similar to or greater than those seen in Microraptor and Anchiornis though there is no clear pattern relating body size to speed, as the largest (London) and smallest (Eichstatt) specimens yielded similar values (Table S8). Only Microraptor and all specimens of Archaeopteryx showed the ability to achieve takeoff velocities by this method alone (Table S8).

Figure 3 Evolution of flight stroke enhancements to flap running (orange) and vertical leaping (blue) performance.

Estimated ranges are mapped over a phylogeny of selected Maniraptoriformes. Averages are presented when multiple specimens are available. Upper lines are for 90° flap angles and lower lines for 50° flap angles. Flight-stroke specific characters are mapped onto the phylogeny: 1, forelimb integument; 2, pennaceous feathers on forelimb; L-shaped scapulocoracoid; 4, laterally facing glenoid; 5, asymmetrical remigies; 6, alula; 7, incipient ligament-based shoulder stabilization; 8, dorsolaterally facing glenoid; 9, full ligament-based shoulder stabilization. The bottom coloured lines denote 50° flap angles and upper coloured lines 90°. Silhouettes from PhyloPic images by B. McFeeters, T.M. Keesey, M. Martynuick, and original.

Leaping

The use of forelimbs during jumping was divided into three discrete analyses, one examining the potential of the wings to increase maximum jump height, one to examine distance gained horizontally, and finally to see if the wings could generate enough force to take off from a standing start as seen in most extant birds.

Vertical

No non-paravian gained more than 8% additional height with flapping using the highest flap angles, and most gained less than 3% (Fig. 3, Table S9). Using more reasonable flap angles of 50°, none exceeded 4%. Within paravians, several taxa generated greater than 10% height increases, including Anchiornis, Microraptor, Eosinopteryx, Changyuraptor, Aurornis and all Archaeopteryx specimens (Table S9). Despite this most troodontids, both the “short armed” Jehol Dromaeosaurs, Mahakala and Sinornithosaurus showed values more similar to non-paravians, between 1–8.5% increase in height. Of interest, the “four winged” taxa used here (Anchiornis, Microraptor, and Changyuraptor) yielded increased height gains on the order of 16–64%, with Microraptor gui specimens showing values in excess of 50% (Fig. 3, Table S9). Even under the lowest flap angle settings, both specimens of M. gui showed leaping height increases of greater than 30%, almost four times the value for the non-paravians under any setting, and Changyuraptor and Microraptor hanqingi showed values of approximately 20%, which is greater than twice the highest value seen in any non-paravian. All Archaeopteryx specimens showed height gains greater than 30% under all mass permutations, with the lighter estimates for the Berlin, Thermopolis and 11th specimen exceeding 190% non-flapping height values. Interestingly the only specimen that did not reach the 50% height gain under any permutation is the Eichstatt specimen, the smallest in our analysis, whose range between 34–48% gains is similar to what is seen in the larger microraptorine specimens (excluding Sinornithosaurus).

Horizontal

Similar to vertical leaping, there was a marked disparity between distance gained in the “four winged” paravian taxa and all others (Table S10). Only one non-paravian Similicaudipteryx STM-22, under the highest setting and at a 45° takeoff angle, showed distance increases of 5% or greater. Among paravians Microraptor, Changyuraptor, the smaller Anchiornis and all species of Archaeopteryx show leaping values greater than 20% non-flapping horizontal distance at the 45° take off, though this drops to 15% at 30°.

Vertical takeoff

Among non-avians, only Microraptor gui achieved body weight supports greater than 1 under any flap angle or flapping frequency permutation under the two avian derived take off speeds assessed. No non-paravian showed values greater than 0.15 bw under these conditions (Tables S11–S13). Outside of Microraptor, Changyuraptor and the smaller specimen of Anchiornis, deinonychosaurians did not have values beyond 0.5 bw under either speed or any flap frequency permutation. In avians at the lower body weight estimate, all taxa showed values greater than 1.0 bw at the high end of their flapping angle range. At the higher mass estimates, multiple specimens of Archaeopteryx showed levels below 1.0 bw, with the lowest values seen in the Eichstatt and London specimens (Tables S11–S13). Many extant avians use launch speeds between 1.5 m/s and 3.8 m/s (Earls, 2000; Berg & Biewener, 2010; Heers, Dial & Tobalske, 2014). At these takeoff speeds avians more derived than Archaeopteryx achieved values in excess of 1.0 bw, with the exception of the larger mass estimates of Sapeornis under the ALL and GF flapping estimates (Tables S4–S6 and S11–S13). At the higher speed of 5.1 m/s, achievable by strong leapers, beyond Microraptor the only other non-avian theropods to achieve greater than 1.0 bw support was the smaller specimen of Anchiornis under a single flap rate permutation at 90° flap angle.

Discussion

A major challenge of attempting to create models that examine evolutionary transitions is that of efficiency versus effectiveness. Evolved traits may need to only function at some basic level, rather than contribute high degrees of functional adaptation. Thus, an argument against our use of thresholds, such as a 6% body weight support as the minimum for WAIR, is that smaller values, such as 5% or even 1%, may still provide selective advantages for individuals. Although this line of thought is defensible, we suggest a challenge to this. The first is that these low values are not testable in the sense that there are not physically defined thresholds to demarcate when a behaviour may or may not function. Without these parameters to test, any discussion becomes a story-telling scenario. In addition, we have used liberal parameters in reconstructing extinct taxa based on output values measured in modern, derived avians. This optimistic reconstruction of the possible ignores that non-avian theropods have additional functional restrictions based in their musculoskeletal, neuromuscular and integumentary systems not present in extant birds. The minimal age of origin for powered flight in avian theropods where is 130 million years ago (Wang et al., 2015) and this behavior and all its functional and morphological components have been under refinement through selection ever since. Thus, we postulate that the claim that non-avian theropod would be able to perform functions at output levels below the threshold minimums seen in extant avian taxa difficult to defend. For example, flapping frequency and flap angle have large effects on the resulting body weight support values and using avian take off values are likely significant over estimations for values obtainable in most if not all the taxa sampled here. Our use of a velocity of 1.5 m/s is based on the speed of adult Chukars, whose WAIR ability is much greater than proposed of any non-avian taxa examined here. Using juvenile values (0.6 m/s of stage I) reduces the bw support values by approximately one third. Additionally, by using coefficient of lift values of 1, which is higher than is seen in a 20 dph Chukar at 45° angle of attack (stage II per Jackson, Segre & Dial, 2009), we are likely highly positively biasing the results. Thus, we argue that due to our relaxed constraints and the significantly higher wing loadings to that seen in any stage of Chukar development (even the asymmetrical crawling stage of 1–3 dph from Jackson, Segre & Dial, 2009), the taxa sampled here that did not reach the 0.06 bw threshold derived from in vivo experiments or meet the wing loading values seen in the earliest stages of ontogeny should not be considered WAIR capable. Although we do not have in vivo derived values to compare with leaping and flap running estimates, it is not parsimonious to propose that small incremental increases measured only under unnaturally lenient conditions support a behavior.

For all behaviours tested here there is a sharp contrast in performance levels between a small number of paravian taxa (Microraptor, Anchiornis, Changyuraptor, Aurornis and Eosinopteryx) and all other non-avian taxa. This discrepancy is marked not only because it does not correlate to the origin of pennaceous feathers at pennaraptora but it also does not include all members of Paraves within the high performing category. Multiple small bodied and basal members of both deinonychosaurian subgroups, such as Mahakala, Xiaotingia, Jinfengopteryx, Mei, Sinovenator and Sinornithosaurus, show little evidence of benefit from flapping assisted locomotion. As these taxa are similar in size to the paravians that do show potential benefits, the argument that this loss is a byproduct of allometry is not possible. Allometric loss of performance is possible though in the larger, feathered dromaeosaurs like Velociraptor (∼15 kg, Turner et al., 2007) or Dakotaraptor (∼350 kg, Depalma et al., 2015), but our data from embryonic maniraptorans does not support this postulate. As our measurements for the small paravian wing areas are based either on preserved feather length (Sinornithosaurus) or on long feathered close relatives (Anchiornis for Xiaotingia, Jinfengopteryx, Mei, Sinovenator and Microraptor for Mahakala) our values for them are likely overestimates and suggests that locomotion was not a major driver for forelimb evolution, even among small sized paravians.

Flap running

There are questions as to whether a flap running model is particularly efficient for any taxa. One immediate set of constraints relates to performance of the hind limb under a potential flap-running model. The thrust production model we used assumes the hindlimb and forelimb propulsion potentials were simply additive. However, in reality the hindlimb performance must have some maximum output that is likely to be exceeded if the forelimbs produce significant additional propulsive force. Thus, at high wing-produced thrust production, the hindlimbs likely cannot move fast enough to accommodate the faster speeds. Under such conditions, an animal would pitch forward and fall.

We also assume that most of the lift produced by the wings during flap-running could be oriented as thrust. The proportion of force that can be oriented as thrust is, however, constrained by wing kinematics, particularly the amount of spanwise twist that the wing can undergo during the flight stroke (Iosilevskii, 2014). Thus, our thrust proportions for theropods may be unrealistically high, overestimating the speed generated.

Additionally, downstroke lift production not reoriented as thrust would act to displace some weight. Although this is important and necessary in flight, it would reduce hindlimb performance during flap-running by reducing the normal force acting through the feet. A similar phenomena occurs during high angled WAIR (Bundle & Dial, 2003). Finally, the production of lift during flap-running, regardless of orientation relative to travel, would generate significant amounts of drag (including profile drag, pressure drag, and induced drag). Given these potential performance constraints, it is questionable whether flap-running would be as effective a locomotion mode as our data suggests, even for taxa like Microraptor.

WAIR

The finding that not a single non-paravian reaches the 6% bodyweight threshold for level 1 WAIR challenges the proposal that WAIR offers a behavioural pathway for basal maniraptorans (Dial, Randall & Dial, 2006; Heers, Tobalske & Dial, 2011; Heers, Dial & Tobalske, 2014). The few cases that approach these values (Similicaudipteryx, Yulong, and Yixianosaurus) are only achieved under wing angle and wing beat permutations that are unrealistic given their pectoral musculoskeletal structures (Baier, Gatesy & Jenkins, 2007; Turner, Makovicky & Norell, 2012). MOD derived wing beat values in beats per second for the larger Similicaudipteryx (6 Hz), Yixianosaurus (7–8 Hz), Yulong (10 Hz) are greater than or equal to those of smaller extant birds such as the Magpie (Pica pica) (9.2 Hz), Crow (Corvus brachyrhynchos) (6.6 Hz) and Raven (Corvus corvax) (6.1 Hz) (Jackson, 2009) and are so elevated due to the inclusion in that dataset of galliform birds, which are short burst specialists with shortened wings, large pectoralis and supracoracoideus muscle masses and muscle fiber adaptations to maximize their flight style (Askew & Marsh, 2001; Tobalske et al., 2003). These specialized muscles are adapted to allow wing beat frequencies beyond those of other birds at a similar body mass (Tobalske & Dial, 2000; Tobalske & Dial, 2007; Jackson, 2009; Jackson, Segre & Dial, 2009) thus inflating our wing beat frequency estimates. Wing beat frequencies were likely much lower in non-avian theropods than in modern birds during takeoff, which is higher than during level flight (Dial, 1992; Berg & Biewener, 2010), given the relatively small size of their wing musculature and plesiomorphic musculoskeletal anatomy (Jasinoski, Russell & Currie, 2006; Allen et al., 2013; Baier, Gatesy & Jenkins, 2007; Bock, 2013; Burch, 2014).

In none of our nine permutations did values indicating level 1 WAIR performances become unambiguously optimized at Paraves (Data S1). This is despite our conservative application of constraints such as use of a 90° flap angle, flap frequencies comparable of greater than many extant avians, WAIR velocity comparable to adult Chukars and generous wing area estimates. In paravians that do shown positive scores, these are no more than 0.12 bw under 90° flap angle at a velocity of 1.5 m/s and any flapping frequency reconstruction for the larger Anchiornis, Aurornis, Eosinopteryx or Sinornithosaurus and Changyuraptor under all but the MOD flapping rate estimate (Table 3). This suggests that tightening these constraints either singularly or combination would likely exclude marginally performing taxa from even this threshold. For example, using the body velocity of 6–8 dph Chukars (0.6 m/s) at 70° flap angle, excludes Aurornis, the larger Anchiornis, Eosinopteryx under all permutations and Changyuraptor except under the MOD flapping frequency.

Given the low values seen Aurornis and reduced flapping ability in Eosinopteryx (Godefroit et al., 2013) it is likely that only the juvenile Anchiornis specimen, Microraptor and Changyuraptor among non-avian theropods would even have the potential to use this behavior. When we introduce other factors in addition to those listed above such as the symmetrical feathers or the plesiomorphic pectoral girdle would likely have limited the prevalence of WAIR further, if present at all, to only the microraptorines as they would have further reduce the effectiveness of the wings in force generation. Feather asymmetry aids in resisting out of plane forces and is crucial for their bending and twisting during the flight stroke (Ennos, Hickson & Roberts, 1995; Norberg, 2002). While the pectoral girdle morphology of Anchiornis which show non-elongated and convex coracoid and lack of ossified sternum or fused gastralia, denote reduced pectoral muscle mass compared to microraptorines (Zheng et al., 2014). This does not make a strong case that this behavior was present ancestrally in Paravians, yet alone that it coincided with pennaceous feather evolution and elongation (present at Pennaraptora) or other flight related adaptations. Our findings suggest that if present at all, there is a minimum of two origins for the use of flap-based locomotion with the presently accepted phylogenetic hypotheses; once within microraptorines, and once in Aves. This is not completely surprising, as other traits related to flight, such as an alula, elongated coracoid, and a broad, ossified single sternum plate, are also independently derived in Microraptor and basal avians that are more derived than Sapeornis, suggesting convergent evolution in early powered flight (Zheng et al., 2014).

To compare the results of our body mass and wing area estimates to others in the literature we ran the WAIR and leaping takeoff analyses using previously published mass and wing area values for Archaeopteryx (Yalden, 1984), Microraptor (Chatterjee & Templin, 2007; Alexander et al., 2010), Caudipteryx and Protarchaeopteryx (Nudds & Dyke, 2009). In all cases, WAIR values were similar, often below, values calculated in our analysis (Table S14). Non-paravians yielded WAIR values near 0 bw and take off speeds were required to be greater than 46 m/s. Microraptor specimens showed takeoff velocities between 4.1–6.6 m/s, values achievable either by running or leaping methods and similar to those estimated in our original analysis.

Locomotory pathways to flight: necessity or red herring?

Our first principles modeling approach, which accurately predicts WAIR values for Chukar chicks, supports the postulate that for these “near flight” behaviors, wing area is the major determinant of function rather than power. One potential argument for why a locomotory pathway is required for the evolution of flight related characters is that the muscle hypertrophy in the pectoral girdle present in extant flying birds would be unlikely to have evolved passively if display or stability flapping methods drove the origin of large wings. Although it is undeniable that extant avians have proportionally and significantly more wing musculature than non-avian theropods, the minimum level needed to achieve a ground-based takeoff is unknown. There are several volant birds with flight muscle ratios (flight muscle mass proportion of total mass) below 16% (Marden, 1987). Juvenile Chukars that fledge less than two weeks after hatching (Harper, Harry & Bailey, 1958; Christensen, 1970; Christensen, 1996) and young peafowl (which fledge after one to two weeks Fowler, 2011) also have ratios below this value. Recent estimates for Microraptor yield values within this range (Allen et al., 2013).

Fledging aged Chukars and Peafowl have a reduced flight muscle fraction compared to adult birds. In Chukar’s, at 14–15 dph, the pectoral mass is only 48–62% the relative size (as a proportion of total mass) compared to adult birds, while in Peafowl (12 dph) this range is between 38–45% (Heers & Dial, 2015). Yet at this age the wing loading values are significantly less than in adults, with 15 dph Chukars showing values only 38% of adults and 11–14 dph Peafowl showing values ranging from 22–25% of those seen in adults. Among non-avian theropods only Microraptor (specimens BMNHC PH 881, IVPP V 13352, LVH 0026 under Alexander et al., 2010’s mass estimate) and the juvenile Anchiornis (BMNHCPH828) have similar wing loading values to fledging aged Chukar (10–17 dph) (Heers & Dial, 2015). Of these, only Microraptor and early avians have previously been suggested to have similar pectoral muscle mass fractions (pectoral limbs region 13–15% of total mass per Allen et al., 2013) combined with similar wing loading values as seen in volant juvenile Chukars (minimum forelimb muscle mass of 14% of body mass, wing loading values below 80 N/m2). Thus, we contend that these taxa may have had a power output that would be capable of ground based take off, as the reduced pectoral musculature was compensated for by their large wing size.

Even at slight lower estimates of flight muscle, mass percentage take off may be possible in Microraptor and basal avians. Early fledgling aged Chukar chicks show forelimb muscle mass fractions (Heers & Dial, 2015) below the 16% suggested as the minimum for takeoff by Marden (1987). This is due to their proportionally large wings. With such a proportionally large wing area, even at low forelimb mass fledging aged Chukars can that generate lift values estimated at between 10.4–12.2 N/kg of body mass (using the muscle-specific power output value of 360 W/kg per Askew, Marsh & Ellington, 2001) which exceeds the minimum needed for takeoff (9.8 N/kg) (Marden, 1994). Therefore, if wing area can partially overcome the need for significant muscle mass fractions arguments on the need for a selective pathway to muscle hypertrophy need not be invoked when discussing the origins of flight. This would also help explain the lack of features indicating significant hypertrophy in pectoral musculature, such as a lack of a sternal plate, in the earliest fliers (Zheng et al., 2014) and the delayed presence of a keel until Ornithothoraces (O’Connor & Zhou, 2015). These findings suggest that powered flight originated before pronounced muscle hypertrophy and likely depended more on wing loading and shoulder mobility. Thus, the pathway to large pectoral muscles is one that occurred within Aves, not before and likely is linked to the refinement and extension of level flight capabilities.

For WAIR, a similar tradeoff between muscle mass and wing area likely exists. In juvenile galliforms, flight muscle mass increases logistically throughout ontogeny. In Chukars this goes from about 2% in crawling, non-WAIR capable 3 dph juveniles to 26–29% in 100 + dph adults (Heers & Dial, 2015). Individuals capable of stage I WAIR (8 dph, maximum WAIR angle 65°) have proportional muscle masses between 7.5–9.9% of body mass, which represents 25–40% of adult proportional pectoral mass values (Heers & Dial, 2015). They also show wing loading values only 55–60% those of an adult, which should be noted can achieve much larger maximum WAIR angles (> 90°). A similar pattern is seen in both late Stage II WAIR Chukars and in juvenile Peafowl. The former can ascend up to 85° despite showing reduced pectoral muscle mass relative to body mass (48–62% adult values) but have wing loading values only 40% those of adult birds. Juvenile peafowl, which at 12 dph can achieve higher WAIR angles than adults, display less than half the relative pectoral muscle mass fraction of adults, but have wing loading value of only 1/4 to 1/3 that seen in adults (Heers & Dial, 2015). This suggests that reducing wing loading could partially compensate for the lower proportional muscle mass, an idea that is also supported by findings in Brush Turkeys where low wing loaded juveniles can WAIR whereas adults cannot (Dial & Jackson, 2011).

We generated a model for Chukar WAIR ontogeny that predicts wing loading, pectoral mass, maximum WAIR angle, and age using data from Heers & Dial (2015) (Fig. 4). Most relationships are nonlinear and multimodal, suggesting complex interactions between these factors. The original and modeled data show an inflection point between 20–30 dph. Up to this age, maximum WAIR angle asymptotes at less than 90° (Jackson, Segre & Dial, 2009; Heers & Dial, 2015). This corresponds to when the pectoral muscles reaches ∼20% total body mass and the beginning of Stage III where both extended level flight and vertical flight is possible (Jackson, Segre & Dial, 2009). Here is also when we begin to see, through in vivo measurements, the steady increase in wing loading values from their minimum of 55 N/m2 at day 22 continuing upwards to the full term (100 + dph) score of 161 N/m2.

Figure 4 3D scatterplot of values for Chukars modeled for the first 70 days of growth.

2D projections of the values are shown on each axis-pair plane with grey circles. Age, pectoral limb muscle mass, wing loading, and WAIR performance data are from Heers & Dial (2015). Maximum WAIR angle was limited to 100°. Regressions were neither linear nor unimodal suggesting a complex interaction between musculoskeleletal and aerofoil ontogeny and performance. Mass (g) was estimated from age by the quadratic equation 5.730818 + 3.472647 × x + −0.011605 × x2 + 0.000661 × x3 (R2 = 0.9902); only ages less than 100 days were used. Percent pectoral mass was estimated from mass by the quadratic equation 0.858022 + 0.231592 × x −0.000658 × x2 × 5.9340−7 × x3 (R2 = 0.92). Wing loading was estimated from mass by the quadratic equation 1.692164 + −0.018717 × x + 8.756264−5 × x2 + −9.483335−8 × x3 (R2 = 0.69). Maximum WAIR angle was estimated from mass by the quadratic equation 38.119489 + 1.137820 × x + −0.007969 × x2 + 1.925223e − 05 × x3 (R2 = 0.9575).

Early stage Chukar chicks have forelimb masses within the range suspected for non-avian theropods (up to 15 dph) and we see a correlation among these chicks between maximum WAIR angle and lower wing loading (Figs. 4 and 5). WAIR capable Chukar chicks during this period, which corresponds to late Stage I through Stage II of Jackson, Segre & Dial (2009), show relatively constant wing beat frequencies (22–26 Hz) and flap angles (∼140°) further supporting the idea that wing loading is a major factor influencing maximum WAIR angle. Wing loading values in WAIR capable galliforms are significantly below that seen in much of our dataset and only eight specimens, pertaining to five paravian taxa show wing loading values below 200 N/m2 (Table 1; Fig. 5). Of these, only Microraptor, a juvenile Anchiornis, and Eosinopteryx show wing loadings that, according to this model, suggest WAIR is even possible. Given that the flapping frequencies and stoke angles under those seen in the extant Chukars for which this relationship between this compensatory mechanism for low muscle mass occurs, the levels they achieve are likely beyond non-avian theropods. This suggests that this compensatory pathway would likely be less efficient or even unavailable to most non-avian theropods, again likely restricting WAIR potential to only the microraptorines.

Figure 5 Regression of measured wing loading versus maximum.

WAIR angle in Chukar chicks aged 3–15 day post hatching and estimates for selected non-avian theropods. Chuckar data are from Heers & Dial (2015). Large circles denote Chukar values with their age given as the number inside. Regression for Chuckar data is 100.17 − 20.824x, R2 = 0.848. Small circles denote estimated paravian theropods. Only specimens with wing loading values comparable to those seen in Chukars (< 2.0 g/cm2 = 196 N/m2) were included. Demarcation of quadrupedal crawling to WAIR at 65° was based on Jackson, Segre & Dial (2009). Non-avian theropods are: f1, Anchiornis huxleyi BMNHCPH828; f2, Anchiornis huxleyi LPM B00169; f3, Aurornis xui YFGP-T5198; f4, Changyuanraptor yangi HG B016; f5, Eosinopteryx brevipenna YFGP-T5197; f6, Microraptor gui BMNHC PH 881; f7, M. gui IVPP V 13352; f8, M. hanqingi LVH 0026 (light mass estimate); f9, M. hanqingi LVH 0026 (heavy mass estimate).

Our first principles modeling approach, which accurately predicts WAIR values for Chukar chicks, supports the postulate that for these “near flight” behaviors, wing area is the major determinant of function rather than power. Many possible selective regimes can be put forward for driving the expansion of wing area before it would provide any locomotory benefit. These include display (Hopp & Orsen, 2004; Zelenitsky et al., 2012), egg shielding (Carey & Adams, 2001), braking, or balance (Fowler et al., 2011), and our results suggest that they need to be investigated in greater detail in order to understand the drivers for major pre-requisites for the flight stroke and reduced wing loading. The flight stroke itself, once we have divorced it from the early expansion of the wing and the origin of pennaceous feathers, likely occurred after expansion into the wing-loading region where wing based locomotory regimes are possible. Thereafter, multiple possible scenarios can be sought to explain the origin of flight stroke and flight itself, with potentially different scenarios occurring in different lineages. Our data indicates that, whichever scenario, WAIR would be restricted in its functional presence to, at the earliest, small-bodied Paraves or more likely the base of Aves; well after previous suggestions (Heers & Dial, 2012).

Ontogenetic versus phylogenetic signals

The findings of our model that all non-paravian theropods and most deinonychosaurians were incapable of using WAIR, raises the question of when along the lineage could WAIR have evolved and under what selective context? As our data shows there is no evidence of WAIR in non-paravian theropods, this challenges the hypothesis that modern bird ontogeny recapitulates the pathway to the origin of flight. Although it is tempting to suppose that behaviours young, non-volant extant birds undertake can offer some insight into the origins of flight, modern bird chicks do not present plesiomorphic morphologies. Although extant birds hatch with somewhat reduced forelimb muscle masses and feathering, the musculoskeletal morphology is still generally comparable with adult extant fliers. For example, near-hatchling quail embryos do not have an ossified sternal keel, but instead have a cartilaginous or connective tissue based on (Meneely & Wyttenbach, 1989; Tahara & Larsson, 2013; Fig. 5). Some birds, such as chickens, which are bred for greatly enlarged pectoral muscles, do develop a broad sternum with a robust midline keel in ovo (Hall & Herring, 1990). In most non-avian theropods, including many small paravians, the sternum is either composed of a pair of unfused plates or completely absent (Xu, Wang & Wu, 1999; Hwang et al., 2002; Gong et al., 2012; Godefroit et al., 2013; Zheng et al., 2014; Lü & Brusatte, 2015) with the notable exception of Microraptor gui (Xu et al., 2003), thus it is unlikely to have even a cartilaginous or rudimentary keel seen in juvenile birds. Beyond this the oblique acrocoracohumeral ligament orientation and triosseal canal and a dorsally oriented glenoid fossa are also present in extant avian embryos, even in poor fliers like Chukars, but not in non-avian theropods. These differences combined with those in muscle mass and neuromuscular pathways differentiate the ontogentic transitions of juvenile birds from evolutionary ones regarding avian origins. This is especially true as the exemplar non-avian theropod taxa (Dial, Randall & Dial, 2006; Heers & Dial, 2012; Heers & Dial, 2015) do not represent an anagenic sequence but are instead derived members of lineages separated by tens of millions of years.

Modified flapping behaviors are present in other birds that can’t fly, such as steaming in pre-fledgling ducklings (Aigeldinger & Fish, 1995), begging and signaling in altricial chicks (Rydén & Bengtsson, 1980; Glassey & Forbes, 2002; Ngoenjun & Sitasuwan, 2009), and social displays and thermoregulation in Ostriches (Bolwig, 1973; Mushi, Binta & Lumba, 2008). This indicates that even in the most basal lineage of extant avians, the ancestral flight stroke has been modified by juvenile and non-volant individuals to perform other tasks. Even late stage avian embryos and wingless hatchlings perform coordinated flapping motions on their own and when stimulated (Hamburger & Oppenheim, 1967; Provine, 1979; Provine, 1981a; Provine, 1981b; Provine, 1982) showing that the neurological pathway for flapping motion is active and functioning before hatching in precocial birds (Provine, 1979). These embryonically established neural controls are thus available to the earliest hatchlings of modern birds (volant or not) but non-avian theropods may not have had neuromuscular control or the coordinated flapping behaviours even extant chicks do.

Although ontogenetic trajectories are relatively linear, with regards to a species, phylogenetic trajectories are not. The WAIR capabilities of extant birds may be a direct result of their advanced powered flight adaptations rather than a precursor for it. Because the factors that facilitate WAIR are the same as those that permit flight (increased wing area, muscle resources, and flapping arc), WAIR may be more of a spandrel that extant birds have capitalized on rather than a selective pathway. Thus, we propose instead that juvenile birds exapted the flight stroke for use as an escape technique before they were capable of takeoff and flight, and this derived escape response was only possible once the complex flight adaptations of derived birds evolved.

Ground takeoff

Although no thrust based locomotory method succeeded in providing an adequate evolutionary pathway with an obvious evolutionary trend that surpassed biophysical thresholds, some individual specimens did succeed at crossing these thresholds under certain parameters. Notably, Microraptor gui and Archaeopteryx showed significant results in all three methods. Interestingly, both taxa were estimated to have had the potential for ground based takeoff at both sprint speeds and leaping takeoff values (Tables S8 and S11–S13). Given the effects of flap running’s thrust generation (though see potential limitations below), takeoff speeds can be achieved with a starting velocity well within the range of similar sized extant tetrapods. Even a sprint speed, without wing assistance, of 7 m/s is not unrealistic given greater speeds are obtained by the Roadrunner (Lockwood, 2010), Red legged Seriemas (Abourachid, Höfling & Renous, 2005), multiple small mammals (Iriarte-Díaz, 2002), and some lizards (Huey, 1982; Clemente, Thompson & Withers, 2009).

Living birds that launch by running are overwhelmingly aquatic or semi-aquatic taxa, suggesting that running takeoff is mostly an adaptation to compliant surfaces (as referenced in Earls (2000)). Other birds utilize a leaping takeoff to initiate flight with high instantaneous speeds during leaping (Biewener, 2003), easily matching the values used here. The required speed values for takeoff we calculated could be lowered if we assumed a coefficient of lift above 1.5, similar to those seen during takeoff in extant birds (Usherwood, 2009) or if we reduced our mass estimates. Microraptor has an elongated hindlimb, especially when compared to basal birds of similar snout-vent length (Dececchi & Larsson, 2013). These proportionately longer hindlimbs may have not only increased top running speed, as leg length is related to stride length and speed (Garland & Janis, 1993; Hoyt, Wickler & Cogger, 2000), but also leads to an overestimation of body mass because body masses for theropods are generally derived from femur length (Dececchi & Larsson, 2013). If we reduce the mass of Microraptor gui (IVPP V 13352) to that of a similar sized Archaeopteryx specimen (Solnhofen) we get a mass estimate of between 0.4–0.6 kg, or between 42–67% of the value used here for IVPP V 13352. This is similar to differences we see between mass estimates of femur length and 3D models for LVH 0026 (Tables S1 and S14). Using 0.6 kg for Microraptor, values greater than 1.0 bw are achieved at speeds of only 3.8 m/s, and even less if Cl values closer to extant birds of 1.64 are used. This suggests that at reasonable speeds, even with a coefficient of lift below that of extant birds, Microraptor was likely capable of ground based take off. Also during leaping take off, the horizontal velocity of birds increases rapidly after the first few strokes (Berg & Biewener, 2010). Therefore, effective flight strokes coupled with a strong ability to jump would supply ample velocity to help achieve vertical takeoff.

Although no single locomotory behaviour tested here surpasses minimal thresholds for high incline running or powered flight, a flight stroke in stem avians may have had performance benefits to biomechanical scenarios that are more difficult to test. Specifically, feathered forelimbs, coupled with a nascent flight stroke, may have contributed subtle, but evolutionarily advantageous performance benefits to high speed maneuvering and braking and balancing during prey capture. Even slight performance enhancements to vertical and horizontal leaping may have had highly positive adaptive effects. Enhancements of even a few percent may had tremendous advantages to these animals, particularly if we compare the small margins of performance differences of extant predator-prey interactions. Unlike leaping, WAIR is a behavior with minimal thresholds that must be overcome. As such incremental gains cannot be achieved until that threshold is reached, something that we find, despite our relaxed conditions, is not present in the majority of non-avian theropods and may have been restricted solely to the microraptorines and avians. Thus, the hypothesis that incremental gains in WAIR would have adaptive benefits and drove forelimb and pectoral evolution in non-avian theropods is not supported as no non-paravian maniraptoran show any capability to perform this behavior.

Conclusion

All models tested here suggest that the feathered forelimbs of all non-paravian theropods and most non-avian theropods were not capable of surpassing the minimal physical thresholds of powered flight and WAIR. The origin of pennaceous feathers was not tied to a dramatic locomotory shift in these early non-avian theropods. Non-paravian taxa such as Caudipteryx, Similicaudipteryx, and Yixianosaurus have forelimb feathers greater than 100 mm in length, and similar sized feathers are suspected on other oviraptorosaurs (Paul, 2002; Hopp & Orsen, 2004), large dromaeosaurs (Depalma et al. 2015) and even ornithomimids (Zelenitsky et al., 2012; van der Reest, Wolfe & Currie, 2016). These structures represent a significant energetic investment for structures that we estimate to have had minimal locomotory benefits. Moreover, the symmetry of the vanes of the pennaceous feathers in these taxa would make the feathers aeroelastically unstable, further constraining their use in a locomotor context (even the pennaceous feathers of microraptorines may have been somewhat unstable during aerial locomotion, with vane asymmetries below the critical values for functional aeroelastic flutter reduction see Feo, Field & Prum (2015)). These taxa also possessed large tail feathers that were likely used for display (Pittman et al., 2013; Persons, Currie & Norell, 2014) and feather melanin based pigmentation likely coincides with the origin of pennaceous feathers (Li et al., 2010; Li et al., 2014). This suggests other non-locomotory functions such as display or brooding were likely significant evolutionary driver for pennaceous feather growth (Hopp & Orsen, 2004; Zelenitsky et al., 2012).

The mosaic evolution of flight related characters suggests the evolution of the flight stroke was not continuous in this clade, nor driven by a single overall driver. If different behavioural traits or selective regimes and not a single locomotory function were driving the evolution of feather elongation, one may not expect the concordance of “pre-flight” characters in different coelurosaur clades or even in all members of a single clade. This would explain the non-uniform distribution of traits such as the elongated forelimbs with well-developed feathers (Dececchi & Larsson, 2013; Godefroit et al., 2013; Foth, Tischlinger & Rauhut, 2014), laterally facing glenoid (Gao et al., 2012), and an ossified sternum for muscle attachment (Zheng et al., 2014).

Although it is beyond the scope of this paper to speculate on which driver or combination of drivers led to feather elongation and forelimb musculoskeletal evolution for powered flight, we suggest that future research not focus on any single event or “pathway” to attempt to explain pre-avian evolution of characters later exapted into the flight apparatus. Given the time between the Paravian-avian split and the appearance of the Jehol microraptorines is approximately 40 million years, estimated from the oldest known paravian Anchiornis (161 Ma) and Microraptor (120 Ma) (Xu, Zhou & Wang, 2000; Xu et al., 2009) a single continuous locomotory based evolutionary driver is unlikely. Moreover, it seems unparsimonious to argue that refining flapping based locomotion was central to the evolution of maniraptorans when the lineages show marked difference in their ecology, body size, limb usage and feather extent.

Although the selective pressures for each of these traits is unknown, what is apparent is it that pennaceous feathers and other critical characters related to the evolution of powered flight were not originally adapted for significantly different locomotion. It is also clear that WAIR was not a major driver for the evolution for much of Maniraptora or even Paraves. These findings reshape how we view the origins of birds and the evolution of different maniraptoran clades and refocus our investigations to look at taxa not as steps of a ladder towards the origin of flight, but as organisms adapting to the unique demands of their immediate behavioural and ecological surroundings.

Supplemental Information

Supplemental Information 1 Theropod mapping.

Mapping of W.A.I.R. values across theropod and early avian phylogeny. Topology based on Dececchi & Larsson (2013).

Click here for additional data file.

Supplemental Information 2 Extant avian nexus.

Nexus file for the Modified flapping rate regression. Nodal dates form Jetz et al. (2012). Taxa and measurements from Askew, Marsh & Ellington (2001) and Jackson (2009).

Click here for additional data file.

Supplemental Information 3 Calculations.

Spreadsheet for WAIR and leaping height calculations.

Click here for additional data file.

Supplemental Information 4 Equation description and justifications.

Explanation for equations used.

Click here for additional data file.

Supplemental Information 5 Measurement data.

Measurement data for non-avian and avian theropods used in this analysis.

Click here for additional data file.

Supplemental Information 6 Humerus percentage of forelimb.

Humerus percentage of forelimb calculation compared to bodysize in avian and non-avian theropods.

Click here for additional data file.

Supplemental Information 7 Body width estimation.

Furcula width for Deinonychosaurians and early avians used to calculate body width estimate.

Click here for additional data file.

Supplemental Information 8 WAIR calculation using ALL flapping rate.

WAIR values using flap rate from regression from Jackson (2009) all taxa.

Click here for additional data file.

Supplemental Information 9 WAIR calculation using GF flapping rate.

WAIR values using flap rate from regression of ground foraging birds from Jackson (2009).

Click here for additional data file.

Supplemental Information 10 WAIR calculation using MOD flapping rate.

WAIR values using flap rate from regression based on modified dataset adding galliform birds from Askew, Marsh & Ellington (2001) and Jackson (2009).

Click here for additional data file.

Supplemental Information 11 Flap running.

Increased in velocity after 10 iterations for flap running analysis.

Click here for additional data file.

Supplemental Information 12 Modelling passerine bird take off.

Take off calculations for passerine birds from Jackson (2009).

Click here for additional data file.

Supplemental Information 13 Vertical jumping.

Height gain due to flap based thrust for non-avian theropods and Archaeopteryx.

Click here for additional data file.

Supplemental Information 14 HOrixontal jumping.

Horizontal distance gain due to flap based thrust for non-avian theropods and Archaeopteryx.

Click here for additional data file.

Supplemental Information 15 Leaping take off values using ALL.

Body weight support values for ground based take off with a leaping speeds of 3.8, 4.1 and 5.1 m/s using flap rate from regression from Jackson (2009) all taxa.

Click here for additional data file.

Supplemental Information 16 Leaping take off values using GF.

Body weight support values for ground based take off with a leaping speeds of 3.8, 4.1 and 5.1 m/s using flap rate from regression of ground foraging birds from Jackson (2009).

Click here for additional data file.

Supplemental Information 17 Leaping take off values using MOD.

Body weight support values for ground based take off with a leaping speeds of 3.8, 4.1 and 5.1 m/s using flap rate from regression based on modified dataset adding galliform birds form Askew, Marsh & Ellington (2001) and Jackson (2009).

Click here for additional data file.

Supplemental Information 18 WAIR and leaping takeoff based on previous models of Microraptor, Archaeopteryx, Caudipteryx and Protoarchaeopteryx.

WAIR and leaping takeoff values for models taken from the literature. Data for Archaeopteryx from Yalden (1984), Microraptor specimens from: Chatterjee & Templin (2007), Alexander et al. (2010) and Dyke et al. (2013). Caudipteryx and Protoarchaeopteryx from Nudds & Dyke (2009).

Click here for additional data file.

Additional Information and Declarations

Competing Interests

Author Contributions

Data Deposition

The authors declare that they have no competing interests.

T. Alexander Dececchi conceived and designed the experiments, performed the experiments, analyzed the data, contributed reagents/materials/analysis tools, wrote the paper, prepared figures and/or tables, reviewed drafts of the paper.

Hans C.E. Larsson contributed reagents/materials/analysis tools, wrote the paper, prepared figures and/or tables, reviewed drafts of the paper.

Michael B. Habib conceived and designed the experiments, performed the experiments, contributed reagents/materials/analysis tools, wrote the paper, reviewed drafts of the paper.

The following information was supplied regarding data availability:

Dryad: 10.5061/dryad.1f5h4.

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
