# Peer review of "The wings before the bird: an evaluation of flapping-based locomotory hypotheses in bird antecedents"

_PeerJ, doi:10.7717/peerj.2159_

## Round 0.1 · original submission · Major Revisions

Overall, the paper is on that cusp between minor and major revisions, but the items are just extensive enough that I have labeled it as "major revisions". That said, my reading of the reviews is that none of the issues fundamentally sink the paper, and I am confident that you will be able to address them in revision.

Among the comments from the reviewers (all of which should either be incorporated into the revision itself or addressed in the response letter), I wanted to particularly highlight the following:

- For Figure 1, you will need to check and ensure that all of the images parts are released under a CC-BY license, or that you have permission to do so (particularly for the Jackson figure components--these may need to be removed or redrawn to be acceptable under a CC-BY publishing license). If any of the images are CC-BY-SA or CC-BY-SA-NC, you will need additional permissions from the artists.

- In agreement with the reviewers, I would strongly recommend that Data S4 (the equation description and justifications) be moved into the main body of the paper. These are fairly critical to understanding the paper and its methods, and don't take up that much space. One of the reviewers also notes some potential errors in Data S3, which should be fixed.

- The reviewers provide some important clarifications and corrections for many of the citations within the text, and these should be addressed in revision.

- I note that a few comments were left on the preprint version of this paper, and you should feel free to deal with those in the revision as you best see fit.

·

Basic reporting

The writing of this manuscript needs to be improved before publishing. Some issues are smaller, such as clauses and sentences where the subjects are not clear, but there are some larger issues. There are two paragraphs in particular that I can't sufficiently review because I do not understand them at this time. The first is the discussion starting at line 687 where two evolutionary scenarios are discussed. Presently, I do not understand the distinction between them that the authors are attempting to draw. The second is the opening paragraph of the "Energetics and WAIR" section. It is not clear how the evidence provided leads to the conclusion drawn in the last statement. I've annotated other writing issues in the pdf I have attached to this review, but I think the authors should also perform a thorough read-through of the entire manuscript themselves.

Experimental design

No comments

Validity of the findings

No comments

Additional comments

An interesting addition to the dialogue about the origin of flight, I think that the paper would be a worthwhile addition to the literature if some concerns are addressed. The first has been touched on in the previous section, which is the writing. The second concern is that there are a number of mistakes in "Data_S3_Calculations", detailed below. A reader attempting to replicate results by using this information, rather than the excel files, would not be able to. A third concern is that few results are in the main paper. Most results are in the supplemental tables. With the focus on WAIR, I recommend adding a table with those results into the main body of the text. It would not have to be as complete as the supplemental tables; columns for taxon, amplitude, and body weight should cover the most important information.

Another concern is that some of the equations are described as modifications to existing equations for pterosaur launch, and then a reference to Witton and Habib 2010 is given. As far as I can tell, the equations are not given in that paper. Right now, this could be read as intending to imply that the equations were published in that paper.

I suggest the authors consider including an explicit discussion of assumptions and limitations of their method in the paper's discussion. This would not have to be anything lengthy, but I think that directly addressing potential criticisms helps frame this sort of paper.



Following are specific comments not covered in the annotated .pdf that has additional comments:

Table 1 and others: "Microrpator"

Data_S3_Calculations:
Body weight support calculations: The equation here "Amp= f b" is incorrect. In the excel sheet, b is multiplied by the stroke angle in radians, not the frequency, which makes more sense. Side note: this gives the arc length rather than the chord length which is generous in that it should overestimate the amplitude, but this should be noted.

"For calculations of leaping take off they seen in extant birds 3.8m/s (starlings) and 4.1 m/s (quails) from Earls (2000) as well as that from Galagos (Gunther et al. 1991). " Sentence fragment.

"A=Vbal tlaunch
Ballistic velocity (Vbal= Dun tlaunch)" - both missing a division sign?

"Power required=MVbalAg" - doesn't seem right, the power should be M*V*A. Looking at the excel sheet it looks like this should be (A+g)

Figure 3: This graph is currently confusing. Performance gain should be defined in the caption as this specific term is not used in the text. The silouhettes which I think are indicating flight capability are not explained. The offsetting of the axes of flap running and leaping versus wair can work, but the labeling needs to be better. Probably using left and right sides for the two different axes would be a good start.

Table S10- Gain seems to be measured differently here than in S9. Switch this measurement to match S9; having everything start at >100% is harder to parse.

Tables S11-13: Need labels to differentiate them from each other

Reviewer 2 ·

Basic reporting

See comments in general regarding writing and figures.

Experimental design

No Comments

Validity of the findings

There are potentially significant areas to be addressed regarding data use and conclusions. Details in general comments.

Additional comments

The authors apply data derived from extant taxa, that have been hypothesized to be relevant to the question of the evolutionary origins of bird flight, to models of fossil animal performance. They use functional data to bracket potential performance metrics (for example leap height or body weight support) of the numerous extinct taxa they analyze. They conclude that WAIR and flap-running would not have reached adaptively beneficial levels phylogenetically until well after penaceous (or “flight”) feathers evolved, concluding that hypotheses that the flight stroke in aerodynamically functional behaviors drove the avian flight bauplan early in their evolution are false. This is a timely and important analysis. The approach the authors take is powerful, and given their data, some of their conclusions are warranted. However, I note below significant issues with their data and interpretations. I also found many typos, writing inconsistencies, and confusingly worded sentences that made the paper difficult to follow, especially for a more general audience. Some of my confusion, I think, stems from the fact some important information is included only in the supplemental material. For example, it’s unclear in the main text how they are using velocity (see my comment below), but the supplemental describes how it’s used.


line 86 - 89: This is a fundamental failure in interpreting the literature - the paper cited as Dial and Jackson 2010 regards Australian Brush Turkeys, and describes a reverse ontogeny of performance.
Based on Tobalske and Dial 2007 (6-8 day old chukars), and Heers et al 2011 (8-12 day old chukars) climbing 65-75° using WAIR requires aerodynamic forces of ca. 10% of body weight. It’s not until >80° that forces exceed the stated 50% (to 60%). And adult chukars (which don’t approach the listed 0.8kg) can run up inverted obstacles, rather than struggle (as stated). The authors should also see Heers & Dial 2015 (evolution) regarding large developing peafowl (see supplemental data) .

L 101 - need a year in the citation

L105-107: The function of flap-running in water fowl is unknown. In fact, it is just as likely that the hindlimb are not assisting with horizontal acceleration (as written), but with weight support until the speed needed to generate weigh-supporting lift can be achieved. And the Earls 2000 cite is questionable, and very similar to the relevant line in that discussion. Nothing in that paper discusses waterfowl save for this one line: “Second, a running take-off of the type proposed in evolutionary hypotheses is seen in select groups of living birds that are morphologically specialized (e.g. albatrosses, loons), that are taking off from highly compliant surfaces (e.g. water) or are that are very large (e.g. swans).” This idea is also misattributed to Earls in line 768.

L117 “forelimb and hindlimb”, the listed citations differ in their arguments about whether both forelimb and hindlimb (or just forelimb) pennaceous feathers “aided locomotion”

L134 - “…force generations were estimated using powers calculated from a 10% and 30% proportion of body mass…” I’m unclear what this means. ’m assuming the means estimating muscle mass as 10-30%, but is this just flight muscle mass or total body muscle mass? How does one estimate force using powers?

L135-137 - This needs to be explained. If Allen et al.’s numbers did not include M. pectoralis, how are they relevant for estimating flight muscle power?

L153-155 - Confusingly written sentence.

L248 - There was an error in the Bundle and Dial force plate data regarding GRF; the authors should be sure to read the supplemental material in Dial and Jackson, 2009.

L246-249: Based on Jackson et al 2009 (and contrary to the author’s statements here, but like what they say in the introduction regarding crawling), stage I chukars barely perform WAIR but can ascent ca. 60°; WAIR doesn’t fully kick in until 65° and stage II individuals, which can ascend 75° producing small (10-20% bw) forces. From an adaptationist perspective, the important thing that the authors should consider is that for these bipeds, 10% is a functional cut-off, because that’s the difference between <60° and (at a minimum) ca. 70°.

L 321 - Velocities: is this movement of COM?

L321-345: When the authors say “we chose” for each of these variables, what exactly are they choosing? Are these model inputs? Or are they performance benchmarks to be “successful”? It’s still under the subsection of “wing range of motion” so the purpose of this paragraph is unclear. After reading the supplemental material, I’ve answered my question, but the main text should be clarified so that the supplemental material is not necessary for a basic understanding of the procedure.

A second point regarding velocities, running speed in general is positively correlated with leg length (or hip height), so the extant-based numbers used in the model should be adjusted for estimated leg length, or the lack of inclusion should be justified.

L367-368: a very confusing sentence.

L383-407: Confounded in this discussion is the relationship between the high wing loading in the chukars and the ontogenetic stage regarding development of the neuromuscular system (as discussed in Jackson et al, 2009). The authors here should consider comparing their numbers to Brush Turkeys (Dial and Jackson 2011), which increase in wing loading with age while decreasing in WAIR performance; and since the hatchlings are capable of flight, this comparison would remove the minimal neuromuscular development seen in the early stage chukars.

L601-615. What are wing-beat values? Those numbers in parentheses. Are these frequencies? What are the units?

L630 - the role of symmetrical feathers regarding aerodynamics is questionable. Other than leading edge feathers, most extant remiges and retrices are minimally asymmetric.

L665 - a very confusingly written sentence

L702 and on - Two comments here. First, as long as flapping behaviors are within the anaerobic capabilities of the animal, and the anaerobically derived duration of the behavior would provide some adaptive benefit, then the energetics of the behavior is practically irrelevant. Vertical burst take off in birds, is extremely costly, but they can do it, so they do. Without quantification of the energetics of extant birds in any of the mentioned behaviors, this is an untestable discussion.
Second, while the authors use frequency and amplitude as indices to compare energetics (muscle work and/or power), muscle energetics is actually dependent on force production, which the authors neglect. Force production in flight muscles is sensitive to wing inertia, wing area, and angle of attack.
In other words, given both points above, estimating the energetics of these behaviors is difficult in extant taxa, and untestable in extinct.

L749 - The authors state that chukar flight morphology in chicks is “comparable” to adults, and that chicken embryos “develop a broad sternum with a robust midline keel”. Based on Heers and Dial 2012 (TREE, fig. 2), chukar chicks capable of ascending steep angles do not have either a robust keel, or a flight morphology comparable to the adults. Chickens have been artificially selected for fast growth of reproductive systems or meat muscles, including m. pectoralis, hence we might expect them to develop a keel at an earlier age than most birds. 

L769 - “Absence of a keeled sternum in stem avians implies that these embryonic  rhythmic pectoral contractions were also absent.”
What is the support for this statement? The keel, as the authors state, is relevant to muscle hypertrophy - presumably force production. Rhythmic movements do not necessarily imply high force production and bone remodelling/adaptation.

L784-786 - The end of this sentence is ambiguous. “…some taxa did.” Did what?


L 823-827 - I couldn’t agree more that slight improvements in performance may have had highly adaptive benefits. But that goes for being the predator (as implied here), or the prey flap-running (or WAIR’ing) away from a non-winged predator.

L840 - “…the symmetry of the vanes of the pennaceous feathers in these taxa would make the feathers aeroelastically unstable”. Again, the asymmetry of feathers is only aerodynamically important in leading edge feathers. For example, Jackson et al 2009 show wing drawings that show symmetrical feathers posterior to the leading edge in young birds (stage I and II) capable of producing sufficient aerodynamic forces for WAIRing.

L862 paragraph - A very agreeable idea that a single solution to such a complex pathway (or likely set of pathways) is unlikely

Fig. 1 - this is a copy of the figure from Jackson et al 2009 with a few new data points added for the current data. However, in copying the figure, the authors neglected the copy the radial axis label, and I had to look up what the numbers mean (age in days). So how then did the authors place the data points, given that they have neither age nor WAIR angle for the extinct taxa?

Fig. 2 - As mentioned above, the authors should consider using Brush turkey wing loading for comparison since it involves fewer confounding factors of neuromuscular development - but a very cool trend shown here.

---

## Round 0.2 · Minor Revisions

After a second round of review, the reviewers have provided a handful of final comments. Please give these close attention during your revision.

·

Basic reporting

The writing in the results and discussion is still rough in places, and in some cases edits seem to have introduced new errors. I would recommend making use of a copyeditor or a colleague willing to read carefully. At the least I would recommend the authors give the manuscript a slow and careful read through before final submission.

Experimental design

No comments

Validity of the findings

No comments

Additional comments

Some supplemental files appear not to have been updated correctly. For instance, S5 still has "microrpator" and S10 appears unchanged. It is not clear whether the revised files were not uploaded, or if they were not properly added to review materials. The authors should ensure the correct versions have been uploaded.

Otherwise the authors have addressed my previous comments satisfactorily. There are a few remaining issues I think should be dealt with before publication, including the writing, but nothing substantial enough to require another review.

Following are specific comments:

356: I see where the confusion is coming from. Part of the first paragraph on range of motion reads "Abduction of the forelimb beyond 90 deg from the ventral vertical plane was not possible in most non-avian theropods." When I read the sentence at 356 I read it as referring to the same 90 deg- as an angular description of the forelimb in space i.e. maximum abduction. I would just clarify what is meant e.g. "Rotating/moving/etc. through the full 90 deg is likely unattainable for all non-avian theropods due to the constraints of the substrate and shoulder angle..." I would also get rid of "angle" in substrate angle, because it sounds like there is a relationship between incline steepness on wing excursion.

411: later used twice, while I think the authors want "former" and "latter"

435: Why are some values given in N/m^2 and some in kg/m^2?

469-472: Still confusingly written. The first clause sets up a group: "non-avians or archaeopteryx" that does not get higher scores than .33. The second clause says that only microraptor and anchiornis score >= .1 . What happened to archaeopteryx?

476: Why the shift to % body weight for birds, while non-avian dinosaurs were proportion of body weight?


Figure 2 caption- missing the number 3. Pair of lines described twice, keep the first description.
Figure 3- missing the number 3. Pair of lines described twice, again. "Estimated ranges are mapped over a phylogeny of selected Maniraptoriformes"- I don't think so.

Reviewer 2 ·

Basic reporting

There are still a number of unclear or ambiguous statements detailed below.

Experimental design

No Comments

Validity of the findings

I have one other general concern focussed in the discussion regarding wording. The authors make a number of statements regarding what extinct taxa could or could not have done. For example on L846: “The findings that all non-paravian theropods and most deinonychosaurians were incapable of using WAIR” is speculation that is supported by their model estimates. Many of their conclusions properly refer to “based on our model” or “using these permutations”, but sentences like the one on L846 make a claim that no amount of data, save a time machine, could back up.

Additional comments

The resubmitted manuscript has been improved, but I still find a number of problems. First, there remain a number of typos, spelling mistakes, grammatical errors, inconsistencies with abbreviations, very difficult to follow sentences, and other problems, some of which made the review difficult. I highlight a few below, but I urge the authors and the editor to take a very close look. I do not only see such errors as a writing issue, but some can indicate a lack of attention to detail that hopefully does not also permeate through the calculations. I checked several of the spreadsheets and did not find immediate errors, but I urge the authors to double check. Regarding the confusing sentences, many could be clarified by the inclusion or deletion of commas, and/or splitting the sentence into two. I also think that the methods area needs to be restructured - though it is improved over the first submission. The entire set of results is based on several equations which are only available in the supplemental materials. I think the equations (or at a bare minimum the body weight force calculation: Bw=0.5Cl*p *(fAmp +U)2 S/9.8M), with brief descriptions of the included variables, should be in the main paper, so that a reader may fully understand (but not necessarily replicate) the research without viewing the supplemental files. This is a problem across many journals now forcing methods out of the main manuscript.

Other than those problems, I still think this is a valuable analysis and a good addition to the ongoing discussion. The authors make several important points (e.g. efficiency vs efficacy) that have been missing from some of the literature. My hope is that the authors continue this line to model the effectiveness of other proposed hypothesis for the evolution of avian flight (e.g. flapping descent vs. gliding) using similar techniques.

Detailed comments below:

Comments on Response to Reviewers:
Regarding line 86-89 (now line ~115): I see my mis-interpretation of the writing. Since the authors only mention chukars, and then later mention “larger birds” without mentioning other species, I was concerned that the authors were still discussing chukars, which would be inaccurate. I suggest they add mention to other species (brush turkeys and peafowl) to reduce the risk of the confusion I experienced. In regards to coefficients of friction, certainly sandpaper has a higher coefficient of friction than lumber, but that is irrelevant to natural conditions. Tree bark of most species is highly irregular and coarse, and natural incline substrates could include any number of rocky structures as well, both of which could provide far more traction than sandpaper. Though that distinction and discussion has minimal influence on the results.

Regarding symmetrical vs asymmetrical feathers: I do not disagree with the comments. However, there is a history in the paleontological literature of simple statements that symmetrical feathers could not have been aerodynamically functional without qualification; I do not want the authors to fall into that camp. Symmetrical feathers might be unstable if orientated towards perpendicularly to air flow, but that certainly does not preclude an aerodynamic function given other orientations or arrangements.

Specific comments on the resubmitted manuscript:

L 99: “escape into trees (Dial 2003)”, most of the relevant papers, including the ones cited, use terminology similar to “elevated substrates” or “inclined substrates” that might include boulders, cliffs, and/or trees. I suggest changing “trees” to something more inclusive.

L 164: “tot”

L176: “pervious”

L188-191: Sentence should be split into two for readability

L245-249: This is one of the very awkward sentences and is very difficult to read. Plus, it is missing punctuation or includes extra in several places.

L280 and througout. There are inconsistencies with abbreviations. For example, a space or not between the number and “bw”; or Cl vs CL for coefficient of lift.

L314 (and throughout). Chukars is misspelled several times, including here, in the text, figure legends, and in the supplementary material. Plus, scientific names should be given for all included species.

L325: “reported in the literature”. Need citation(s). The next sentence says a “close match”. How close? Where is the comparison? If it’s in a table, please include a table reference.

L350: run on sentence - need a comma after the “)”.

L357 “aligned with the of the…”

L372: “pre-loading values would be 2.4” Without reading the supplemental materials, there is no indication of what “pre-loading” is, or why 2.4 is important. This is indicative of having too much of the methodology only described in the supplemental material, and why more should be included in the main body of the manuscript. I suggest that the editor provide some guidance regarding what should be included where.

L429: Are the wing loading values in Nm-2 or kgm-2, or g cm-2? Given that the units vary through the manuscript and in the tables, I suggest the authors double check their calculations of wing loading throughout, and use consistent units.

L453: missing a period after “)”.

L466: extra period

L479: sentence fragment

L480: “…there is no clear trend in WAIR capability and allometry.” Please clarify.

L482: should be “All birds *more* derived…”?

L483: “estiamte”, Also, the sentence is very difficult to understand.

L502-505: Confusing sentence.

L585-588: Confusing sentence

L663: should use full common and latin names for each species (also “v” is there instead of “hz”)

L669: “…thus inflating our wing beat estimates.” Should this read “wing beat *frequency* estimates”? Otherwise, what is being estimated regarding the wingbeat?

L667 “…despite our the…”

L693-698: Confusing sentence

L714 “wair values near 0” - I’m assuming this means values of body weight support (or “0 bw”)

L736: pectoral mass is only 48-62%…: is this relative or total? Since muscle mass can be expressed as a percent of body mass, and this is a percent change of a percent (or is it total?), the wording should be clarified as to what is being stated.

L744 “pervious”

L755 “This”. What is “this” referring to? As I read it, the sentence is claiming that a low muscle mass fraction is due to large wings, which doesn’t make sense. I think the authors mean that “take off may be possible in Microraptor…” is possible even at low estimates of muscle mass because of their large wings. But that was two sentences prior. This paragraph needs to be rewritten for clarity.

L757 “10.4-12.2 N/kg”: Is the kg referring to body mass or flight muscle mass? I think it’s body mass, but the 360 W/kg is referring to muscle mass-specific power output, and the 9.8N/kg in the next sentence is referring to body mass specific lift (thought the authors do not clarify any of that).

L774-778: Confusing sentence

L853: a duplicate sentence

L857 and on; and comments in response to referees: Regarding soft tissue keels in some of these young gallinaceous birds (and noting that my knowledge of fossil anatomy is poor) - has a connective tissue or cartilaginous keel been ruled out in the relevant fossils, given that ossified bone is typically the only internal structure fossilized? One might hypothesize that a cartilaginous keel in developing birds is therefore potentially similar to extinct species.

L867 and Figures 2-3: “do not represent an anagenic sequence but are instead derived members of lineages separated *but* tens of millions of years”. Typo in *. Also, given that statement, figures 2-3 need to be modified. I like these figures, and splitting them was very helpful. However, the continuous plot implies a continuous evolution of performance abilities. A box-plot style, not showing connections between taxa, would be more appropriate.


Fig 4 and 5: These are an improvement over the previous figure. Comments: wing loading given as g/cm2. Please check the units given the N vs kg discussion above. I suggest that the authors use consistent units throughout.

Figure 5 and text: The authors mis-state the description of Stage I chukar chicks from Jackson 2009. From the cited source: “(i) Stage I
From 1 to 3 dph, the birds flapped their wings asymme- trically while ascending inclines and the wings were frequently used to crawl quadrupedally up the angled substrate (figure 2 inset, electronic supplementary material video). By 5 dph, flapping became more sym- metric, and the wings rarely contacted the ground. Maximum incline performance did not exceed 65° through stage I (figure 2).

Therefore the demarcation line for quadrupedal crawling should be moved down to line up with the 3 dph data point rather than the 7-8 dph area. In addition, extrapolating the regression line to higher wingloadings than the chukar data is misleading. For example, that specimen (f5 - Eosinopteryx) is listed as being ca. 140 g body mass in table 1. In the WAIR work all birds other than hatchling chukars, including those at similar body masses to Eosinopteryx, are capable of walking or running up 50-60 deg. slopes without using their wings (as the authors state on line 111). If ascending inclines ~ 50-60 deg is possible by bipeds without using wings, then it should be independent of wing loading, and independent of the regression.

Table 2: the column heading for frequency and velocity just give units - they should be wing beat frequency (Hz) and speed (m/s) or similar.


Supplementary materials:

I suggest rewriting the description of the attached spreadsheet named Calculations; it seems to contain sample or example calculations for one specimen. The I interpreted the current sentence, “Please see attached spreadsheet (called Calculations), which includes equations for weight supported during W.A.I.R. / leaping takeoff and for height gain through flap leaping.” as showing all equations.

I don’t know what this is saying:
“This produces an estimate of arc length as opposed to chord length which results in a overestimation of amplitude. “
Does it mean that the equation using wing length, as opposed to one using chord length, estimates arc length; or does it mean that the equation estimates arc length rather than estimating chord length? It doesn’t seem that chord length matters here at all, so reference to it could just be eliminated.

The supplemental equations also describe a calculation of Strouhal number, which, as far as I can see, is never discussed. I suggest eliminating the equation from the supplemental materials.

Missing a parentheses somewhere in the following equation:
Power output= (150 (1 –Muscle anaerobic)+ (400 Muscle anaerobic) Muscle hind

I also suggest the authors include in the supplemental text a table legend containing a brief description of what data each supplementary table contains.

---

## Round 0.3 · accepted · Accept

Thank you for your thorough consideration of the reviewers' comments.

One last minor issue: Note that the spelling for the artist is "B. McFeeters". Also, "PhyloPic".